# Low and contrasting impacts of vegetation $CO_2$ fertilization on global terrestrial runoff over 1982-2010: Accounting for above- and below-ground vegetation-$CO_2$ effects

Yuting Yang[1], Tim R. McVicar[2,3], Dawen Yang[1], Yongqiang Zhang[4], Shilong Piao[5], Shushi Peng[5], Hylke E. Beck[6]

[1] State Key Laboratory of Hydroscience and Engineering, Department of Hydraulic Engineering, Tsinghua University, Beijing, China

[2] CSIRO Land and Water, Black Mountain, Canberra, ACT 2601, Australia

[3] Australian Research Council Centre of Excellence for Climate Extremes, The Australian National University, Canberra, Australia

[4] Key Laboratory of Water Cycle and Related Land Surface Processes, Institute of Geographic Sciences and Natural Resources Research, Chinese Academy of Sciences, Beijing, China

[5] Sino-French Institute for Earth System Science, College of Urban and Environmental Sciences, Peking University, Beijing 100871, China.

[6] Department of Civil and Environmental Engineering, Princeton University, Princeton, New Jersey, USA

*Correspondence to*: Yuting Yang (yuting_yang@tsinghua.edu.cn)

**Abstract.** Elevation in atmospheric carbon dioxide concentration (eCO$_2$) affects vegetation water use, with consequent impacts on terrestrial runoff ($Q$). However, the sign and magnitude of the eCO$_2$ effect on $Q$ are still contentious. This is partly due to eCO$_2$-induced changes in vegetation water use being opposing at the leaf-scale (i.e., water-saving caused by partially stomatal closure) and the canopy-scale (i.e., water-consuming induced by foliage cover increase), leading to highly debated conclusions among existing studies. In addition, none of the existing studies explicitly account for eCO$_2$-induced changes to plant rooting depth that is overwhelmingly found in experimental observations. Here we develop an analytical eco-hydrological framework that includes the effects of eCO$_2$ on plant leaf, canopy density, and rooting characteristics to attribute changes in $Q$ and detect the eCO$_2$ signal on $Q$ via vegetation feedbacks over 1982-2010. Globally, we detect a very small decrease of $Q$ induced by eCO$_2$ during 1982-2010 (-1.7%). Locally, we find a small positive trend ($p<0.01$) in the $Q$-eCO$_2$ response along a resource availability ($\beta$) gradient. Specifically, the $Q$-eCO$_2$ response is found to be negative (i.e., eCO$_2$ reduces $Q$) in low $\beta$ regions (typically dry and/or cold) and gradually changes to a small positive response (i.e., eCO$_2$ increases $Q$) in high $\beta$ areas (typically warm and humid). Our findings suggest a minor role of eCO$_2$ on changes in global $Q$ over 1982-2010, yet highlight that negative $Q$-eCO$_2$ response in semi-arid and arid regions may further reduce the limited water resource there.

## 1 Introduction

Runoff ($Q$) is the flow of water over the Earth's surface, forming streamflow, and representing one of the most important water resources for irrigation, hydropower and other human needs (Oki and Kanae, 2006). Anthropogenic climate change is expected to alter the global hydrological cycle, with greenhouse gas-induced climate warming intensifying the hydrological cycle (Huntington, 2006). Besides climate, terrestrial vegetation also affects the water cycle (Brown et al., 2005). It is well-documented that elevated atmospheric CO$_2$ concentration (eCO$_2$) reduces stomatal opening, which in turn suppresses leaf-level transpiration (Field et al., 1995). If this were the only mechanism that eCO$_2$ changed vegetation this would increase runoff ($Q$) (Gedney et al., 2006). However, eCO$_2$ increases vegetation foliage cover (Donohue et al., 2013; Zhu et al., 2016), leading to enhanced canopy-level transpiration and consequently reductions of $Q$ (Piao et al., 2007). These two opposing responses of

vegetation water use to $eCO_2$ complicate the landscape-scale net effect of $eCO_2$ on $Q$, and existing modeling results are highly debated since they focus on different aspects (i.e., physiological functioning and/or structural change) of how $eCO_2$ affects the plants and thus the water cycle (Fatichi et al., 2016; Gedney et al., 2006; Huntington, 2008; Piao et al., 2007; Yang et al., 2016a; Ukkola et al., 2016b). Moreover, observational and evaluation studies of $eCO_2$ effects on $Q$ remain limited, particularly at regional to global scales.

In addition to stomatal and above-ground vegetation structure responding to $eCO_2$, the below-ground vegetation structure (e.g., rooting depth) is also affected by $eCO_2$, with $eCO_2$ increasing rooting depth overwhelmingly found in experimental observations (Nie et al., 2013) (Supplementary Tables S1 and S2). Deeper rooting depth increases plant-available water storage capacity by allowing vegetation to access deeper soil moisture, which potentially increases transpiration water loss and reduces $Q$, especially during dry spells (Trancoso et al., 2017; Yang et al., 2016b). To date, no previous $eCO_2$-$Q$ modeling attempts have explicitly considered the below-ground $eCO_2$-induced feedback simultaneously with the two previously mentioned above-ground feedbacks: this paper fills that niche.

Here we use a parsimonious, analytical eco-hydrological model based on the Budyko framework (i.e., the Budyko-Choudhury-Porporato, BCP model; Donohue et al., 2012), in combination with an analytical rooting depth model based on ecosystem optimality theory (Guswa, 2008), an analytical $CO_2$ fertilization model for steady-state vegetation (Donohue et al., 2017) and observed plant stomatal response to $eCO_2$ (Ainsworth and Rogers, 2007), to detect the impact of $eCO_2$ on $Q$ changes ($dQ$) via vegetation feedbacks over global vegetated lands for 1982-2010. The Budyko framework describes the steady-state (i.e., mean annual scale) hydrological partitioning as a functional balance between atmospheric water supply (i.e., precipitation, $P$) and demand (i.e., potential evapotranspiration, $E_P$) and a model parameter that modifies the climate-hydrology relationship (Choudhury, 1999; Donohue et al., 2012). In this framework, both $E_P$ and the model parameter are affected by the response of vegetation to $eCO_2$ (see Methods). The 'top-down' (Sivapalan et al., 2003) developed framework allows analytical and transparent attribution of $dQ$ changes, which overcomes the uncertainty raised from non-linear interactions among numerous processes when attributing $dQ$ numerically by using 'bottom-up' earth

system models (Yang et al., 2015). To examine the long-term $eCO_2$ impact and to minimize year-to-year "transient" effects (i.e., water storage changes), we performed our analyses using sequential 5-year periods (Yang et al., 2016a; Han et al., 2020), resulting in six 5-year-means during 1982-2010, with the first period containing 4 years. Additionally, since vegetation response to $eCO_2$ can be greatly mediated by the availability of other resources (e.g., water, light and nutrients) (Donohue et al., 2013; Donohue et al., 2017; Nenami et al., 2003; Yang et al., 2016a; Norby et al., 2010), we examine the impact of $eCO_2$ on $Q$ along a resource availability gradient (Donohue et al., 2017; Friedkubgstein et al., 1999) (see Methods). Resource availability is typically low in dry (and/or cold) environments and increases as the climate becomes more humid, which enables us to detect the signal of $eCO_2$ on $Q$ across a dry – wet gradient.

## 2 Material and methods

### 2.1 Methods

The Budyko-Choudhury-Porporato (BCP) model was adopted here to simulate $Q$ and to attribute changes in $Q$ (Yang et al., 2016b; Donohue et al., 2012). The BCP model uses the Choudhury's (1999) formulation of the Budyko curve to estimate $Q$ (Eq. 1 below), in which the model parameter is estimated based on the relationship between the Choudhury's model parameter and the Porporato's model parameter (Eq. 2 below). The required rooting depth ($Z_r$) in estimating the Porporato's parameter is calculated using the Guswa's (2008) rooting depth model (Eqs. 3-5 below). To quantify the response of $Q$ to $eCO_2$ via vegetation feedbacks, the stomatal response of vegetation to $eCO_2$ is determined by upscaling the observed response at the site level to the biome level (Section 2.1.4) and the Leaf area index ($L$) response to $eCO_2$ is quantified based on the response of water use efficiency ($WUE$) to $eCO_2$ adjusted by the local resource availability following Donohue et al. (2017) (Section 2.1.5). The effects of $eCO_2$ on both stomatal and $L$ also affect rooting depth in Guswa's (2008) model. A flowchart of our modeling approach is summarized in Figure 1 and detailed calculation procedures are described in Sections 2.1.1 to 2.1.5.

## 2.1.1 Runoff simulation

The BCP model adopts Choudhury's (1999) formulation of the Budyko curve, given as:

$$E = \frac{PE_{\mathrm{P}}}{(P^n + E_{\mathrm{P}}^n)^{1/n}} \tag{1}$$

where $E$ is the annual average actual evapotranspiration (mm yr$^{-1}$). $P$ is the annual average precipitation (mm yr$^{-1}$). $E_{\mathrm{P}}$ is the annual average potential evapotranspiration (mm yr$^{-1}$) here estimated using the Shuttleworth-Wallace two-source evapotranspiration model (Shuttleworth and Wallace, 1985; see Section 2.1.2). $n$ is a unitless model parameter that encodes all factors other than mean climate conditions and modifies the partitioning of $P$ between $E$ and $Q$. For assumed steady-state conditions, $Q$ is calculated by subtracting $E$ from $P$ as a result of catchment water balance.

The probabilistic steady-state solution of Porporato's (2004) stochastic dynamic soil moisture model shares a similar form with the Budyko curve (Porporato et al., 2004). Porporato's parameter $\omega$ is a dimensionless parameter, which is a function of effective rooting depth ($Z_r$, mm), mean rainfall intensity ($\alpha$, mm per event) and soil water holding capacity $($WHC, mm$^3$ mm$^{-3}$) and exhibits a close relationship with the Choudhury's parameter $n$ (Yang et al., 2016b; Porporato et al., 2004). A relationship between Porporato's $\omega$ parameter and Choudhury's $n$ parameter was built following three steps. Firstly, we obtained the numerical solution of the Porporato's model of the corresponding $E/P$ for every 0.1 increment in $E_{\mathrm{P}}/P$ for six separate $\omega$ curves. Secondly, by numerically solving Choudhury's formulation of the Budyko curve, we determined the values of Choudhury's parameter ($n$) that correspond to the $E/P$ values of each of the six $\omega$ curves. Thirdly and finally, we pooled all $n - \omega$ pairs together and deduced the relationship between $n$ and $\omega$ ($R^2$=0.96, $p$<0.001; Supplementary Figure S1) as:

$$n = 0.82\ln(\omega) + 0.636 = 0.82\ln(\frac{Z_r \times \mathrm{WHC}}{\alpha}) + 0.636 \tag{2}$$

Effective rooting depth ($Z_r$) was determined using an analytical carbon cost-benefit model based on ecosystem optimality theory proposed by Guswa (2008). The $Z_r$ model is given as:

$$Z_r = \frac{\alpha}{\mathrm{WHC}(1-W)}\ln(X) \tag{3}$$

$$X = \begin{cases} W\left[1 + \dfrac{WHC}{\alpha}\dfrac{(1-W)^2}{2A} - \sqrt{\dfrac{WHC}{\alpha}\dfrac{(1-W)^2}{A} + (\dfrac{WHC}{\alpha}\dfrac{(1-W)^2}{2A})^2}\right] & \text{if} \quad W > 1 \\[4mm] W\left[1 + \dfrac{WHC}{\alpha}\dfrac{(1-W)^2}{2A} + \sqrt{\dfrac{WHC}{\alpha}\dfrac{(1-W)^2}{A} + (\dfrac{WHC}{\alpha}\dfrac{(1-W)^2}{2A})^2}\right] & \text{if} \quad W < 1 \end{cases}$$ (4)

$$A = \frac{\gamma_r \times RLD}{SRL \times WUE} \times \frac{1}{E_{P\_T} \times f_{GS}}$$ (5)

where $W$ is the ratio of the multi-year growing season mean $P$ over potential transpiration, $E_{P\_T}$. $\gamma_r$ is the root respiration rate (g C g$^{-1}$ roots day$^{-1}$), which is quantified using the standard $Q_{10}$ theory (Lloyd and Taylor, 1994; Ryan, 1991) with a fixed $Q_{10}$ coefficient of 2.0 (Zhao et al., 2011). The base respiration
rate at 20 ºC for each biome type is determined following Heinsch (2003). $RLD$ is the root length density (cm roots cm$^{-3}$ soil) and $SRL$ is the specific root length (cm roots g$^{-1}$ roots). We fixed $RLD$ to be 0.1 cm roots cm$^{-3}$ soil and $SRL$ to be 1500 cm roots g$^{-1}$, representing the median value of these two parameters reported in the literature (Caldwell, 1994; Eissenstat, 1997; Fitter and Hay, 2002; Pregitzer et al., 2002). $f_{GS}$ is the fraction of the growing season within a year, with the growing season length
quantified according to Zhu et al. (2016). $WUE$ is the photosynthetic water use efficiency (g C cm$^{-3}$ H$_2$O), which is determined for the first period (i.e., 1982-1985) from the ensemble mean from eight ecosystem models (see Data section) of annual gross primary production ($GPP$) and transpiration ($E_T$) estimates (i.e., $WUE=GPP/E_T$). For the following periods, $WUE$ was estimated by considering the effects of changes in atmospheric CO$_2$ concentration ($C_a$) and vapor pressure deficit ($v$) on $WUE$
(Donohue et al., 2013; Wong et al., 1979; Farquhar et al., 1993) as:

$$WUE_{t+1} = WUE_t + WUE_t(\frac{C_{a,t+1} - C_{a,t}}{C_{a,t}} - \frac{1}{2}\frac{v_{t+1} - v_t}{v_t})$$ (6)

where $t$ is time in year. Note that the above equation implicitly assumes the same upscaling factor when converting the leaf-level assimilation and transpiration to the canopy-level for a given location (Donohue et al., 2017). The spatial pattern of mean annual $Z_r$ is shown in Supplementary Figure S2.

## 2.1.2 The Shuttleworth-Wallace model

The Shuttleworth-Wallace two-source evapotranspiration model (the S-W model) was used to estimate $E_P$ and its two components (potential transpiration, $E_{P\_T}$ and potential evaporation, $E_{P\_S}$) (Shuttleworth and Wallace, 1985). The S-W model estimates evapotranspiration as:

$$\lambda E_P = \lambda E_{P\_T} + \lambda E_{P\_S} = C_T PM_T + C_S PM_S \tag{7}$$

$$PM_T = \frac{\Delta A + (\rho c_p v - \Delta r_a^c A_s)/(r_a^a + r_a^c)}{\Delta + \gamma[1 + r_s^c/(r_a^a + r_a^c)]} \tag{8}$$

$$PM_S = \frac{\Delta A + [\rho c_p v - \Delta r_a^s (A - A_s)]/(r_a^a + r_a^s)}{\Delta + \gamma[1 + r_s^s/(r_a^a + r_s^c)]} \tag{9}$$

$$C_T = [1 + R_c R_a / R_s (R_c + R_a)]^{-1} \tag{10}$$

$$C_S = [1 + R_s R_a / R_c (R_s + R_a)]^{-1} \tag{11}$$

$$R_a = (\Delta + \gamma) r_a^a \tag{12}$$

$$R_s = (\Delta + \gamma) r_a^s + \gamma r_s^s \tag{13}$$

$$R_c = (\Delta + \gamma) r_a^c + \gamma r_s^c \tag{14}$$

where $\lambda$ is the latent heat for vaporization (MJ kg$^{-1}$), $\Delta$ is the gradient of the saturation vapor pressure with respect to temperature (kPa K$^{-1}$), $\rho$ is the air density (kg m$^{-3}$), $c_p$ is the specific heat of air at constant pressure (MJ kg$^{-1}$ K$^{-1}$), $\gamma$ is the psychrometric constant (kPa K$^{-1}$). $r_a^a$, $r_a^c$ and $r_a^s$ are the aerodynamic resistance (s m$^{-1}$) to heat and vapor transfer between the canopy-air space and the atmosphere, between the leaf and the canopy-air space, and between the soil surface and the canopy-air space, respectively. These three aerodynamic resistance terms are estimated following Sánchez et al. (2008). $r_s^s$ and $r_s^c$ are soil surface resistance and stomatal resistance (the reciprocal of stomatal conductance), respectively. To estimate $E_P$ using the S-W model, $r_s^s$ is set to zero and $r_s^c$ is set to its non-water stressed value (Milly and Dunne, 2016). The non-water stressed values of $r_s^c$ for each biome type are provided in Mu et al. (2007). $A$ is the available energy (equals to net radiation minus ground heat flux, W m$^{-2}$) and $A_s$ is the available energy at the soil surface, which is estimated as a function of $L$ following Beer's law (Campbell and Norman, 1998; Yang and Shang, 2013). As a result, $A - A_s$ is the

available energy absorbed by the plant canopy. The impacts of eCO$_2$ on $E_P$ and its two components are obtained by allowing $L$ and $r_s^c$ to vary with $C_a$. Recently, Milly and Dunne (2016) showed that the S-W model could most satisfactorily reproduce evapotranspiration estimates under non-water-limited conditions from climate models under eCO$_2$.

### 2.1.3 Attribution of runoff changes

We used the BCP model to attribute changes in $Q$ ($dQ$) due to different influencing factors following Roderick and Farquhar (2011). To first order, change in $Q$ ($dQ$) is:

$$dQ = \frac{\partial Q}{\partial P} dP + \frac{\partial Q}{\partial E_P} dE_P + \frac{\partial Q}{\partial n} dn \tag{15}$$

where $\partial Q/\partial P$, $\partial Q/\partial E_P$ and $\partial Q/\partial n$ represent the sensitivity of $Q$ to changes in $P$, $E_P$ and $n$, respectively, and can be expressed as:

$$\frac{\partial Q}{\partial P} = 1 - \frac{E}{P}\left(\frac{E_P^{\,n}}{P^n + E_P^{\,n}}\right) \tag{16}$$

$$\frac{\partial Q}{\partial E_P} = -\frac{E}{E_P}\left(\frac{P^n}{P^n + E_P^{\,n}}\right) \tag{17}$$

$$\frac{\partial Q}{\partial n} = -\frac{E}{n}\left[\frac{\ln(P^n + E_P^{\,n})}{n} - \frac{P^n \ln P + E_P^{\,n} \ln E_P}{P^n + E_P^{\,n}}\right] \tag{18}$$

The physiological (stomatal conductance, $g_s$) and structural (Leaf area index, $L$, and effective rooting depth, $Z_r$) parameters impact both $E_P$ and $n$. More specifically, decreases in $g_s$ lower the transpiration rate per leaf area, whereas increases in $L$ and $Z_r$ enhance the canopy-level transpiration rate. Additionally, increases in $L$ also reduce soil evaporation by shading the soil surface (Shuttleworth and Wallace, 1985). The impact of eCO$_2$ on parameter $n$ is expressed through its impact on $Z_r$. On one hand, increases in $WUE$ induced by eCO$_2$ permit a larger vegetation carbon uptake per amount of water loss, potentially leading to more carbon allocated to roots and thus a deeper $Z_r$. Conversely, increases in plant water demand (as quantified by potential transpiration) require vegetation to develop deeper roots to

access deeper soil moisture, and *vice versa* (Guswa, 2008). As a result, we write the functional dependencies of $E_P$ and $Z_r$ as:

$$E_P = f(C_a, E_{P\_M}) \tag{19}$$

$$Z_r = g(C_a, O) \tag{20}$$

where $E_{P\_M}$ is the meteorological component of $E_P$ (without considering the increases in $C_a$). $O$ represents factors other than eCO2 that affect $Z_r$, which effectively encode the climate change-induced vegetation change. With $f$ and $g$ are the functions to describe these relationships. Changes in $E_P$ and $Z_r$ are given by:

$$dE_P = \frac{\partial E_P}{\partial C_a} dC_a + \frac{\partial E_P}{\partial E_{P\_M}} dE_{P\_M} \tag{21}$$

$$dZ_r = \frac{\partial Z_r}{\partial C_a} dC_a + \frac{\partial Z_r}{\partial O} dO \tag{22}$$

Combining Eqs. (2), (15), (21) and (22), we have:

$$dQ = \frac{\partial Q}{\partial P} dP + \left( \frac{\partial Q}{\partial E_P} \frac{\partial E_P}{\partial C_a} + \frac{0.82}{Z_r} \frac{\partial Q}{\partial n} \frac{\partial Z_r}{\partial C_a} \right) dC_a + \frac{\partial Q}{\partial E_P} \frac{\partial E_P}{\partial E_{P\_M}} dE_{P\_M} + \frac{0.82}{\alpha} \frac{\partial Q}{\partial n} d\alpha + \frac{0.82}{Z_r} \frac{\partial Q}{\partial n} \frac{\partial Z_r}{\partial O} O \tag{23}$$

The first term on the right hand of Eq. (23) represents $dQ$ caused by $P$ change and the second term represents $dQ$ caused by eCO2. The third term calculates $dQ$ induced by changes in $E_{P\_M}$ and is calculated as $\frac{\partial Q}{\partial E_P} dE_P - \frac{\partial Q}{\partial E_P} \frac{\partial E_P}{\partial C_a} dC_a$. The fourth and fifth terms on the right hand of Eq. (23) represent $dQ$ caused by changes in rainfall intensity and climate change-induced vegetation change, respectively, and we group them as one factor in the attribution of $dQ$. Since our primary focus was to examine how eCO2 affects vegetation and the consequent impact on $Q$, and its relative importance to changes in $P$ and $E_{P\_M}$, the other factors driving $dQ$ were estimated as the residual of Eq. (23) (i.e., total $dQ$ minus the

sum of $dQ$ induced by $dP$, $dE_{P\_M}$ and $eCO_2$). By introducing Eqs. (17) and (18) into Eq. (23), the sensitivity of $Q$ to $eCO_2$ ($S_{Q\_to\_eCO2}$, mm yr$^{-1}$ ppm$^{-1}$) is written as:

$$S_{Q\_to\_eCO2} = -\frac{E}{E_P}\left(\frac{P^n}{P^n+E_P^{\,n}}\right)\frac{\partial E_P}{\partial C_a} - \frac{E}{n}\frac{0.82}{Z_r}\left[\frac{\ln(P^n+E_P^{\,n})}{n} - \frac{P^n\ln P + E_P^{\,n}\ln E_P}{P^n+E_P^{\,n}}\right]\frac{\partial Z_r}{\partial C_a} \tag{24}$$

The sensitivities of $E_P$ and $Z_r$ to $eCO_2$ (i.e., $\frac{\partial E_P}{\partial C_a}$ and $\frac{\partial Z_r}{\partial C_a}$) are quantified by numerically running the $E_P$ model and $Z_r$ model with and without changes in $C_a$, respectively. The difference between the two simulations under the two $C_a$ scenarios is considered the net effect of $eCO_2$ on $Q$.

### 2.1.4 Stomatal conductance response to $eCO_2$

The response of leaf-level stomatal conductance ($g_s$) response to $eCO_2$ was determined using 244 field experiments with artificially elevated $CO_2$ across a broad range of bioclimates (Ainsworth and Rogers, 2007). We linearly rescaled the reported change in $g_s$ for the magnitude of $eCO_2$ in each of the 244 studies to obtain the sensitivity of $g_s$ to $eCO_2$: that is, the percentage change in $g_s$ per 1% increase in $C_a$. We then classified the 244 observations based on their biome type to construct a biome type-based look-up table of $g_s$ sensitivity to $eCO_2$.

### 2.1.5 Resource availability index and $L$ response to $eCO_2$

The response of $L$ to $eCO_2$ was predicted based on the response of $WUE$ to $eCO_2$ adjusted by the local resource availability. We define a site resource availability index ($\beta$) based on growing season mean $L$ following Donohue et al. (2017). This is because observed $L$ at a site is the net response to the local growing conditions and provides an effective proxy of the growing conditions experienced by vegetation (Donohue et al., 2017). Another advantage of this approach is that $L$ can be readily measured directly or remotely. We calculated $\beta$ as:

$$\beta = 1 - e^{-\tau L} \tag{25}$$

where $\tau$ is an exponential extinction coefficient, which typically varies from 0.3 to 1.2 (Campbell and Norman, 1998) and is set to be 0.7 herein. Broadly across the globe, $\beta$ also corresponds well with climate aridity. The calculated $\beta$ increases from 0.0 with low resource availability (typically dry and/or cold) to 1.0 with high resource availability (typically warm and humid) (Figure 2). This suggests a predominant role of the climate in shaping the global vegetation pattern (Budyko, 1974; Nemani et al., 2003; Yang et al., 2015). This also implies that the resource limitations on plant growth are mainly exerted by climate, consistent with the framework of climate limitation on vegetation proposed in previous studies (Nemani et al., 2003; Budyko, 1974; Yang et al., 2015). Then following Norby and Zak (2011), who showed that the observed response of $L$ to $eCO_2$ was a non-linear function of $L$, we estimated the relative change in $L$ induced by $eCO_2$ per Donohue et al., (2017):

$$\frac{dL}{L} = \frac{dWUE}{WUE}(1-\beta)^2 = (\frac{dC_a}{C_a} - \frac{1}{2}\frac{dv}{v})e^{-2\tau L} \tag{26}$$

## 2.2 Data

The BCP model is validated against observed $Q$ in 2,268 strictly selected unimpaired catchments located across the globe that cover a broad range of bio-climates (Figure 3). Originally, daily and/or monthly $Q$ observations were collected from more than 22,000 catchments globally (Beck et al., 2019). Three selection criteria were implemented to ensure that only catchments with continuous $Q$ records that are negligibly affected by human were used. First, catchments with >5% missing data during the entire study period (1982-2010) were removed. Linear interpolation was applied to fill the gaps in the remaining $Q$ series. Second, catchments smaller than 100 km$^2$ were excluded. This is to ensure that at least one precipitation pixel (i.e., $0.1° \times 0.1°$, or ~100 km$^2$) is included for a catchment. Third, we excluded catchments where observed $Q$ is likely to be affected by human interventions, including catchments with: (i) significant forest gain or loss (> 2% of the total catchment area) (Hansen et al., 2013); (ii) irrigated areas larger than 2% (Siebert et al., 2005); (iii) urban areas (http://ionia.esrin.esa.int) larger than 2%; and (iv) the presence of large dams (Lehner et al., 2011) (i.e., where the reservoir's capacity in a catchment is larger than 10% of the catchment mean annual $Q$). Exactly 2,268 catchments pass these selection criteria (Figure 3).

Precipitation from 1981 through 2010 was sourced from the Multi-Source Weighted-Ensemble Precipitation (MSWEP) version 2 dataset, which has a three-hour temporal resolution and $0.1^\circ$ spatial resolution (Beck et al., 2019). The mean rainfall intensity was calculated as the ratio of annual total precipitation over the number of wet days (with daily precipitation higher than 1 mm; Hartmann et al.,

2013). Other climate variables, including net radiation, air temperature, relative humidity, air pressure and wind speed were obtained from the Multi-scale Synthesis and Terrestrial Model Intercomparison Project (MsTMIP; Wei et al., 2014). To obtain a spatial pattern of *WUE*, global monthly GPP and $E_T$ estimates over 1982-1985 were obtained from eight ecosystem models from MsTMIP (Huntzinger et al., 2013), including: (i) CLM (Mao et al., 2012); (ii) CLM4-VIC (Li et al., 2011); (iii) ISAM (Jain et

al., 1996); (iv) TRIPLEX (Peng et al., 2002); (v) LPJ-wsl (Sitch et al., 2003); (vi) ORCHIDEE-LSCE (Krinner et al., 2005); (vii) SiBCASA (Schaefer et al., 2008); and (viii) VISIT (Ito, 2010). Monthly $C_a$ from 1982-2010 was obtained from the Hawaiian Mauna Loa Observatory (http://www.esrl.noaa.gov/gmd/obop/mlo/) and we assume a uniform $C_a$ concentration across the globe at the mean annual scale (i.e., five years). Monthly $L$ for 1982-2010 was derived from Zhu et al. (2013)

based on AVHRR GIMMS-3g NDVI data (Pinzon and Tucker, 2014). Land cover classification in the year 2001 was acquired from the Moderate Resolution Imaging Spectroradiometer (MODIS) land use map (MOD12Q1) available from the NASA Data Center (Friedl et al., 2010). The global C4 vegetation fraction was obtained from the International Satellite Land Surface Climatology Project (ISLSCP) Initiative II C4 vegetation percentage dataset (Still et al., 2009;

http://webmap.ornl.gov/ogcdown/dataset.jsp?ds_id=932). Soil texture data at $30''$ spatial resolution was acquired from the Harmonized World Soil Database (HWSD) (Nachtergaele, 2009), which was used to determine WHC according to the US Department of Agriculture (USDA) soil classification (Saxton and Rawls, 2006). For catchment scale calculations, these gridded data were further aggregated for individual catchments at a mean annual scale (i.e., five years). For grid-cell analyses, all gridded

datasets were resampled to a $0.5^\circ$ resolution.

# 3 Results

## 3.1 Validation of the BCP model in runoff estimation

The validity of the BCP model is tested by comparing the estimated $Q$ with observed $Q$, in terms of both spatial and temporal variability, at the 2,268 unimpaired catchments (Figure 4). Spatially, the BCP model well captures the observed spatial variability in $Q$ at the mean annual scale, with a coefficient of determination ($R^2$) of 0.93, root-mean-squared error (RMSE) of 87.9 mm yr$^{-1}$ and mean bias (estimated $Q$ minus observed $Q$) of -11.4 mm yr$^{-1}$ (Figure 4a). Temporally, trends in mean annual $Q$ are also reasonably reproduced by the BCP model, having an $R^2$ of 0.71, RMSE of 0.71 mm yr$^{-2}$ and mean bias of -0.05 mm yr$^{-2}$ (Figure 4b). Additionally, we also perform a sensitivity analysis by comparing the simulated $Q$ using the BCP model with and without considering eCO$_2$. Results show that the BCP model, when considering eCO$_2$, performed better in estimating $Q$ trends than the BCP model without considering eCO$_2$, as evidenced by an improvement of $R^2$ by 0.02, a reduction of RMSE by 0.03 mm yr$^{-2}$ and a decrease of mean bias by 0.11 mm yr$^{-2}$, averaged over all 2,268 catchments (Figure 4d). More apparent improvements of the BCP model performance with the consideration of eCO$_2$ are found in regions having a relatively higher resource availability index. For $\beta$ of 0.4-0.6, 0.6-0.8 and 0.8-1.0, the mean bias of simulated $Q$ trends with eCO$_2$ is -0.02 mm yr$^{-2}$, 0.06 mm yr$^{-2}$, -0.36 mm yr$^{-2}$ but increased to 0.24 mm yr$^{-2}$, 0.20 mm yr$^{-2}$ and -0.53 mm yr$^{-2}$, respectively, when eCO$_2$ is not considered (Figure 4d). These results suggest that the analytical framework developed herein captures the eCO$_2$ signal on the observed $Q$ changes.

## 3.2 Plant physiological and structural responses to eCO$_2$

The physiological response of plants to eCO$_2$, that is, the response of $g_s$ to eCO$_2$ is directly compiled from field experiments and summarized for each plant functional type in Ainsworth and Rogers (2007) (also see Supplementary Figure S3). All those field experiments report a reduction of $g_s$ in response to eCO$_2$, with the largest $g_s$ reduction found in C4 crops and lowest in shrubs for the same level of eCO$_2$. On average, for a 1% increase in $C_a$, $g_s$ decreases by 0.47% $\pm$ 0.12% (mean $\pm$ one standard deviation), which means that $g_s$ decreases by 5.67% $\pm$ 1.47% under a 12.1% increase in $C_a$ over 1982-2010 (i.e., from ~343.7 ppm in 1982-1985 to 385.2 ppm in 2006-2010; Keeling et al., 2011). This result is

consistent with a recent isotope-based study (i.e., ~5% reduction of $g_s$ during the past three decades, Frank et al. 2015).

For structural response, averaged across global vegetated lands, our model reveals that elevated $C_a$ has caused an increase of $L$ by 2.12% (0.14% ~ 3.88% for 5% ~ 95% percentile) over 1982-2010 (Figure 5a and b). Despite this relatively small fertilization effect of $eCO_2$ on $L$ at the global scale, an evident gradient is found in the $L$ - $eCO_2$ response that a larger $eCO_2$-induced relative $L$ increase is found in low resource availability regions (smaller $\beta$ value in Figure 2a), and *vice versa* (Figure 5b). This modeled

pattern of $L$ - $eCO_2$ response agrees very well observations at the Free-Air $CO_2$ Enrichment (FACE) observations ($R^2$=0.96, $p$<0.01; Figure 5c) and is also consistent with large-scale satellite-based observations (Donohue et al., 2013; Zhu et al., 2016; Yang et al., 2016a).

In terms of $Z_r$, our modeling results show that elevated $C_a$ over 1982-2010 has resulted in a very minor (0.93%, -0.12% ~ 1.85% for 5% ~ 95% percentile) overall increase of $Z_r$ averaged across the globe

(Figure 5e). Since large-scale observations of $Z_r$ in response to $eCO_2$ are not available, we are not able to quantitatively validate the estimated response of $Z_r$ to $eCO_2$. Nevertheless, the modeled result that $eCO_2$ increases $Z_r$ is overwhelmingly found in site- and/or plant-level experiments (Nie et al., 2013) (Supplementary Tables S1 and S2). Moreover, similar to $L$, the response of $Z_r$ to $eCO_2$ also exhibits a notable difference along the resource availability gradient (Figure 5d and 5e). The positive response of

$Z_r$ to $eCO_2$ is larger in low $\beta$ regions and gradually decreases as the resource availability becomes higher. In high $\beta$ regions (e.g., tropical rainforest and southeast Asia), $Z_r$ even shows a slight decrease in response to $eCO_2$, suggesting a reduced plant water need given the range of $C_a$ over 1982-2010 in those regions.

### 3.3 Attribution of runoff changes over 1982-2010

Over 1982-2010, $C_a$ increased by ~12.1%. For the same period, the BCP model detected a very small reduction in $Q$ of ~1.7% (or 2.2 mm yr$^{-1}$) induced by $eCO_2$ via vegetation feedbacks across the entire global vegetated lands (Figures 6b and 7d). This 1.7% reduction in $Q$, under the context of 12.1% increases in $C_a$, demonstrates a muted response of $Q$ to $eCO_2$. In addition, the overall negative effect of

eCO$_2$ on $Q$ suggests that the structural forcing of eCO$_2$ on vegetation water consumption (both above-

and below-ground) outweighs the physiological effect of eCO$_2$ driving leaf-level water saving. Across

the global vegetated lands and for the same period, the physiological response of vegetation to eCO$_2$ has

led to an increased $Q$ by 0.7% (or 0.9 mm yr$^{-1}$), with the simulated $Q$ increases being increasingly larger

as $\beta$ increases (Figure 6d). By contrast, the structural response of vegetation to eCO$_2$ has resulted in an

overall $Q$ reduction by 2.4% (or 3.1 mm yr$^{-1}$), with the decreases in $Q$ being increasingly smaller as $\beta$

increases (Figure 6e). These two opposite responses of vegetation water use to eCO$_2$ along the resource

availability gradient have led to a significant positive trend ($p<0.01$) in the $Q$-eCO$_2$ response along the

resource availability gradient, from a negative response in low $\beta$ landscapes to a positive response in

high $\beta$ landscapes (Figure 6b). Nevertheless, an exception is found in extreme arid zones (i.e., when

$\beta<0.1$; Figure 6b). This is because in extremely dry areas, the availability of water defines the outcome

and the sensitivity of $Q$ to any changes in land surface properties is very small (Donohue et al., 2013;

Roderick et al., 2014).

We then attribute $dQ$ to different forcing factors between 1982-1985 and 2006-2010 over the global

vegetated lands (Figures 7 and 8). Compared with the early 1980s (i.e., 1982-1985), mean observed $Q$

over the global vegetated lands in the late 2010s (i.e., 2006-2010) increased by 29.7 mm yr$^{-1}$, and the

observed pattern with comparable magnitude in $dQ$ is well captured by the BCP model (Figures 4b and

4d). Consistent with relative $Q$ changes (in %; Figure 6), the impacts eCO$_2$ on the absolute $Q$ change (in

mm yr$^{-1}$) also exhibit a significant upward trend as $\beta$ increases (0.53 mm yr$^{-1}$ per 0.1 increase in $\beta$,

$p<0.01$). Compared to that, increases in $P$ led to a 43.9 mm yr$^{-1}$ increase in $Q$, and enhanced $E_{P\_M}$ has

resulted in a decreased $Q$ by 5.3 mm yr$^{-1}$ (Figure 7f). For the entire vegetated lands and each resource

availability category, the impact of $dP$ on $Q$ generally dominates $dQ$ and is often much higher than that

of eCO$_2$ (Figure 7). An exception is the low $\beta$ regions ($\beta < 0.2$), where the impact of eCO$_2$ on $Q$

outweighs the impact of $dP$ on $Q$ (Figure 8a). As for the impact $E_{P\_M}$ on $Q$, it also shows a notable

gradient with changes in $\beta$ as detected for the eCO$_2$ effect, with the impact of $E_{P\_M}$ on $Q$ being

increasingly negative as $\beta$ increases (Figure 8b-e). The combined influence of other factors including

changes in rainfall intensity (Porporato et al., 2004; Westra et al., 2013) and climate change-induced

vegetation change (e.g., higher $L$) have, in general, exerted a negative impact on $Q$.

Since changes in meteorological factors ($P$ and $E_{P\_M}$) are often considered to dominate changes in $Q$ and have been extensively examined previously (e.g., Roderick and Farquhar, 2011; Yang et al., 2018; Zhang et al., 2018), we next examine the sensitivity of $Q$ to eCO2 ($S_{Q\_to\_eCO2}$) and compare it with the

sensitivity of $Q$ to changes in $P$ and $E_{P\_M}$. Because $C_a$ has different units from $P$ and $E_{P\_M}$, we use relative units to better compare the three sensitivities (Figure 9). Globally, an increase in $C_a$ by 1% only leads to a decrease of $Q$ by ~0.14% (equivalent to ~1.7% for the range of eCO2 experienced over 1982-2010). Similar to the attribution results shown above (Figures 6a and 6b), $S_{Q\_to\_eCO2}$ is generally more negative in global arid ecosystems where $\beta$ is low (Figures 9a and b). The negative $S_{Q\_to\_eCO2}$ diminishes

quickly as $\beta$ increases and becomes positive $S_{Q\_to\_eCO2}$ in high $\beta$ regions. The overall small $S_{Q\_to\_eCO2}$ is further manifested when comparing $S_{Q\_to\_eCO2}$ with the sensitivities of $Q$ to $P$ and $E_{P\_M}$. Averaged across the global vegetated lands, the same relative change in $P$ and $E_P$ would respectively lead to a ~10-times and ~4-times stronger impact on $Q$ than eCO2 does. This highlights the predominant role of climate in shaping the global $Q$ regime (Figure 9c-f and Supplementary Figure S4).

**4. Discussion and concluding remarks**

Elevation in atmospheric $CO_2$ concentration (and other greenhouses gases) is regarded as the ultimate driver of anthropogenic climate change, with consequent impacts on $Q$. Although the impacts of climate change on $Q$ has been extensively studied, the response of $Q$ to eCO2 through vegetation feedbacks is less understood and remains controversial (Gedney et al., 2006; Piao et al., 2007; Huntington, 2008;

Cheng et al., 2014; Trancoso et al., 2017; Yang et al., 2016a; Ukkola et al., 2016a and 2016b). Here, by developing an analytical attribution framework, we detected a very small response of global $Q$ to eCO2-induced changes in vegetation structural (both above- and below-ground) and physiological functioning (Figures 6-8), suggesting that the eCO2 vegetation feedback only exert a minor impact on water resources (partly due to the two opposing water effects between the structural and physiological

responses to eCO2) for the range of eCO2 experienced over 1982-2010.

The overall negative impact of eCO2 on $Q$ detected herein suggests that increased vegetation water consumption driven by the structural response of vegetation (i.e., increases in $L$ and $Z_r$) to eCO2 outweighs the functional change of leaf-level water-saving caused by the physiological effect of eCO2

(i.e., decreases in $g_s$). This result is consistent with previous findings by Cheng et al. (2014), Trancoso et al. (2017) and Ukkola et al. (2016a). In addition, we also detected a significant positive trend ($p<0.01$) in the $Q$-eCO$_2$ response along the resource availability gradient (Figure 6-9). This $Q$-eCO$_2$ response pattern suggests that the structural response of vegetation (i.e., increases in $L$ and $Z_r$) to eCO$_2$ is larger in areas with lower resource availability and gradually decreases as resources become less limiting on plant growth (Figure 5). The positive response of $Q$ to eCO$_2$ in high $\beta$ catchments (primarily located in tropical rainforests; Figure 6a) implies a dominant effect of eCO$_2$-induced partial stomatal closure over increases in $L$ and $Z_r$ on $E$ in these environments (Figure 6). This is reasonable, as both theoretical predictions and *in-situ* observations have consistently reported a negligible response of $L$ to eCO$_2$ in humid and closed-canopy environments (Donohue et al., 2017; Yang et al., 2016a; Norby and Zak, 2011; Körner and Arnone, 1992). In such environments, water is generally abundant with light and/or nutrient availability limiting vegetation growth (Nemani et al., 2003; Yang et al., 2015), and vegetation have evolved to efficiently capture light by maximizing their above-ground structure (i.e., $L$). As a result, in these high $L$ regions, vegetation has already absorbed most of the incident light and any extra leaves would not materially increase the light absorption (Yang et al., 2016a). By contrast, in dry regions, eCO$_2$-induced increase in vegetation water use efficiency (so less transpiration for the same amount of carbon assimilation at the leaf-level) would lead to an increase in $L$ that is directly proportional to an increase in water use efficiency which would increase canopy-level carbon fixation (Figure 5b). This finding is consistent with satellite observations (Donohue et al., 2013) and *in-situ* FACE experiments (Norby and Zak, 2011).

Our findings have important implications for an improved understanding of the global hydrological cycle and managing the world's water resources in a changing climate. Climate models have predicted an increased $Q$ that is primarily driven by an increased $P$ for the 21[st] century (Lian et al., 2021; Milly and Dunne, 2016; Swann et al., 2016; Yang et al., 2018). Here we show that eCO$_2$-induced vegetation feedbacks would mitigate this positive impact of climate change on $Q$ in relatively dry regions and exacerbate the $Q$ increase in relatively wet regions. In addition, higher $C_a$ and increased $P$ enhance the availability of resources for vegetation growth, which increases vegetation coverage or $L$ (Piao et al., 2020; Zhang et al., 2020a; Zhang et al., 2020b). As the vegetation above-ground structural responses to

eCO2 decreases with the increase of $L$, the predicted future $L$ increases suggest that the structural response of vegetation to eCO2 may eventually decrease and the physiological effect of vegetation to eCO2 may become increasingly dominant in the overall response of vegetation water use to eCO2,

leading to an increasing water-saving effect of vegetation in response to eCO2 under future climate change (Zhang et al., 2020b). Analyses of the state-of-the-art climate model outputs already consistently show this water-saving effect of eCO2 globally, especially in relatively warm and humid environments where $L$ is high (Yang et al., 2019). Nevertheless, this may partly be because only some climate models consider the physiological effect while ignoring structural responses of vegetation to eCO2. In addition,

the impacts of eCO2 on $Q$ in relatively dry regions are still highly uncertain and show a great diversity between climate models (Zhang et al., 2020b).

Finally, it is worthwhile noting there are several limitations in the developed modeling framework. First, Guswa's (2008) rooting depth model adopted herein employs an intensive root water uptake strategy, which assumes that root water uptake occurs at a potential rate (i.e., $E_{P\_T}$) until soil moisture

reaches the wilting point when transpiration is completely suppressed (Guswa, 2008). This intensive root water uptake strategy differs from the root water uptake strategy employed in Porporato et al.'s (2004) stochastic soil water balance model, which is a more conservative strategy under which root water uptake linearly decreases with the decrease of soil moisture (Porporato et al., 2004). Combining the two strategies in one modeling framework potentially leads to inconsistency in the theoretical aspect

of the approach. In fact, a later study by Guswa (2010) incorporated Porporato et al.'s (2004) soil water balance model into Guswa's cost-benefit framework for rooting depth (referred to as the Guswa-2010 approach herein). However, the Guswa-2010 approach could not provide an explicit solution for $Z_r$, because the solution of transpiration in Porporato's model is an incomplete gamma function of $Z_r$ (Guswa, 2010; Porporato et al., 2004). As a result, to allow an analytical solution to be derived we used

Guswa (2008) for $Z_r$ in our modeling framework. According to Guswa (2010), using the conservative root water uptake strategy resulted in a slightly deeper $Z_r$ compared to when the intensive strategy was used. Despite that, the response of $Z_r$ to changes in $C_a$ under the two strategies should be similar, as the effects of eCO2 on $Z_r$ are expressed via water use efficiency and $E_{P\_T}$ in our parameterization, which are independent of $Z_r$ parameterizations. This means that adopting different root water uptake strategies

would only lead to differences in the resultant absolute magnitude of runoff ($Q$) but unlikely to result in differences in the response of $Q$ to $eCO_2$, especially when the relative magnitude is used (Figures 5d, 5e and 6a, 6b, 6e and 6f). Alternatively, Rodríguez-Iturbe and Porporato (2004) incorporated the intensive root water uptake strategy into a stochastic soil water balance model and obtained a steady-state solution that has a simper form than Porporato et al.'s (2004) and also mimics the Budyko curve. This

approach deserves further investigation. Second, if interpreted strictly from a theoretical perspective, Porporato et al's (2004) model is more suitable to estimate hydrological partitioning during growing seasons instead of over the entire year as it assumes a constant evaporative demand and precipitation regimes and does not account for snow processes. Expanding all these simplifications, acknowledging imperfect knowledge and parameterisation, would require further analyses to better understand how

they might affect the results shown here. Nevertheless, the uncertainties caused by these simplifications in Porporato et al's (2004) model might be partly overcome during the empirical connection made here between the Porporato's model and the Choudhury's formulation of the Budyko curve, as evidenced by the overall good performance of the developed BCP model in capturing the observed $Q$ (Figure 4). The third limitation of the current study lies in the steady-state assumption of the modeling framework.

More specifically, the steady-state assumption is made in: (i) catchment water balance; and (ii) vegetation functioning. For (i), a five-year period does not necessarily guarantee zero-storage change. Nevertheless, the imbalance in water balance calculation under a steady-state assumption at a five-year scale is generally very small (i.e., typically less than 6% of $P$ in arid regions and less than 3% of $P$ in humid regions) (Han et al., 2020). For (ii), both the Guswa's model for $Z_r$ and Donohue's model for $L$

(see Section 2.1.5) adopted herein were developed for steady-state vegetation (i.e., mature and undisturbed vegetation). Applying these two models to immature (e.g., seedlings) and/or disturbed vegetation can be problematic because immature and/or disturbed vegetation may have very different water use and carbon allocation strategies compared to steady-state vegetation (Donohue et al., 2017; Kuczera, 1987). However, the issues of vegetation age and disturbances are extremely complex and are

well beyond our scope. Moreover, global datasets of vegetation age and disturbances are currently lacking. In this light, our modeled response of $Q$ to $eCO_2$ should be regarded as if all vegetation were mature and undisturbed. Further efforts are needed to better quantify the age and disturbances of

vegetation and to better understand the water use and carbon allocation strategies through the entire vegetation life-cycle and under various types of disturbances.

**Data availability**

All data for this paper are properly cited and referred to in the reference list.

**Author contribution**

YY and TRM designed the study. YY performed the calculation and drafted the manuscript. TRM, DY, YZ, SP, SP, and HEB contributed to results discussion and manuscript writing.

**Competing interests**

The authors declare that they have no conflict of interest.

**Acknowledgments**

This study was supported by the Ministry of Science and Technology of China (Grant No. 2019YFC1510604), the National Natural Science Foundation of China (Grant No. 42071029,
42041004) and the Guoqiang Institute of Tsinghua University (Grant No. 2019GQG1020). T. McVicar acknowledges support from CSIRO Land and Water. The following organizations are thanked for providing observed streamflow data: the United States Geological Survey (USGS), the Global Runoff Data Centre (GRDC), the Brazilian Agência Nacional de Águas, the Water Survey of Canada (WSC), the Australian Bureau of Meteorology (BoM), and the Chilean Center for Climate and Resilience
Research (CR2). Thanks to the HESS Editor and three anonymous reviewers for helpful comments that improved this study.

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

the mean, and the dashed lines represent the 5% and 95% percentiles. The number of grid-cells in each resource availability index category is provided in Figure 2.

**Figure 7** Attribution of changes in $Q$ between 1982-1985 and 2006-2010 across global vegetated lands. **a**, Spatial distribution of changes in $Q$. **b-e**, Spatial distributions of changes in $Q$ induced by (b) changes in $P$, (c) changes in $E_{P\_M}$, (d) eCO$_2$, and (e) changes in other factors (mainly rainfall intensity and climate change-induced vegetation change). **f**, Attribution of changes in $Q$ between 1982-1985 and 2006-2010 averaged over the entire global vegetated lands. Values in the brackets represent one standard deviation of each response among all vegetated grid-cells.

**Figure 8** Attribution of changes in $Q$ between 1982-1985 and 2006-2010 at grid-boxes within each resource availability index ($\beta$) category. Values in the brackets represent one standard deviation of each response among grid-cells within each resource availability index category. The number of grid-cells in each resource availability index category is provided in Figure 2.

**Figure 9** Sensitivity of $Q$ to eCO$_2$ and its relative importance to $P$ and $E_{P\_M}$ across the globe. **a**, Spatial distribution of $Q$ sensitivity to eCO$_2$ (% change in $Q$ per 1% change in $C_a$). **b**, Boxplot of $Q$ sensitivity to eCO$_2$ for each resource availability index category. **c**, Relative importance of eCO$_2$ on $Q$ compared to changes in $P$ on $Q$ (% change in $Q$ per 1% change in $C_a$ compared to % change in $Q$ per 1% change in $P$). **d**, Boxplot of the relative importance of eCO$_2$ on $Q$ compared to changes in $P$ on $Q$ for each resource availability category. **e**, Relative importance of eCO$_2$ on $Q$ compared to changes in $E_{P\_M}$ on $Q$ (% change in $Q$ per 1% change in $C_a$ compared to % change in $Q$ per 1% change in $E_P$). **f**, Boxplot of the relative importance of eCO$_2$ on $Q$ compared to changes in $E_{P\_M}$ on $Q$ for each resource availability category. In **b**, **d** and **f**, the upper / lower box edges represent the quantile divisions, the inner horizontal line is the median, the dots indicate the mean value, and the dashed lines represent the 5% and 95% percentiles. The number of grid-cells in each resource availability index category is provided in Figure 2.

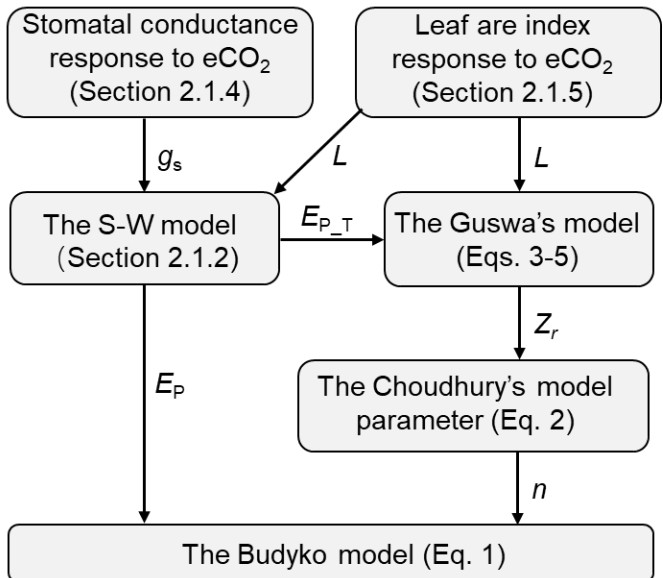

Figure 1 Flowchart of using the analytical models to detect the $eCO_2$ impact on $Q$. The terminologies used are explained in the following text (section 2.1.1 through 2.1.5).

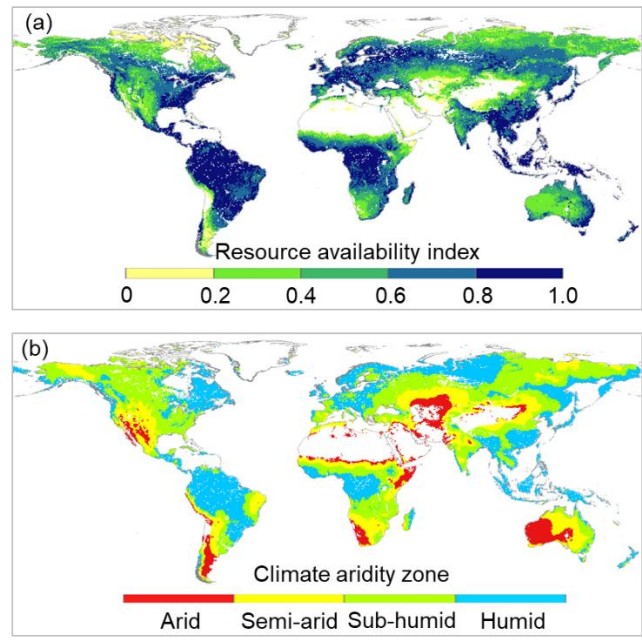

Figure 2 Spatial distributions of (a) resource availability index categories and (b) climate aridity zones over global vegetated lands for 1982-2010. For the land surface blank areas are non-vegetated regions. Respectively there are 2536, 8194, 10316, 12930 and 9093 $0.5° \times 0.5°$ resolution grid-cells in the 0.0-0.2, 0.2-0.4, 0.4-0.6, 0.6-0.8 and 0.8-1.0 resource availability index categories.

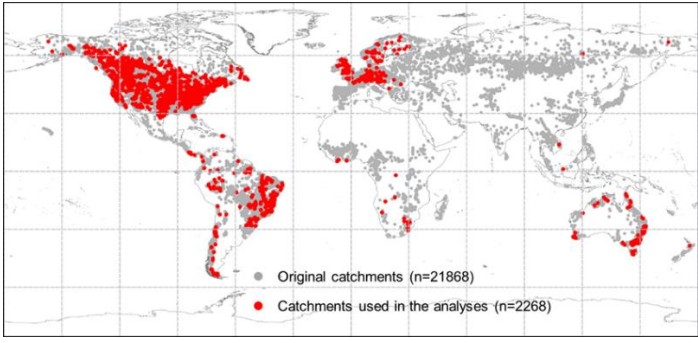

Figure 3 Location of the catchments across the globe. The grey dots show the locations of the original 21,856 catchments, and red dots are the 2,268 catchments that pass the selection criteria and are used herein.

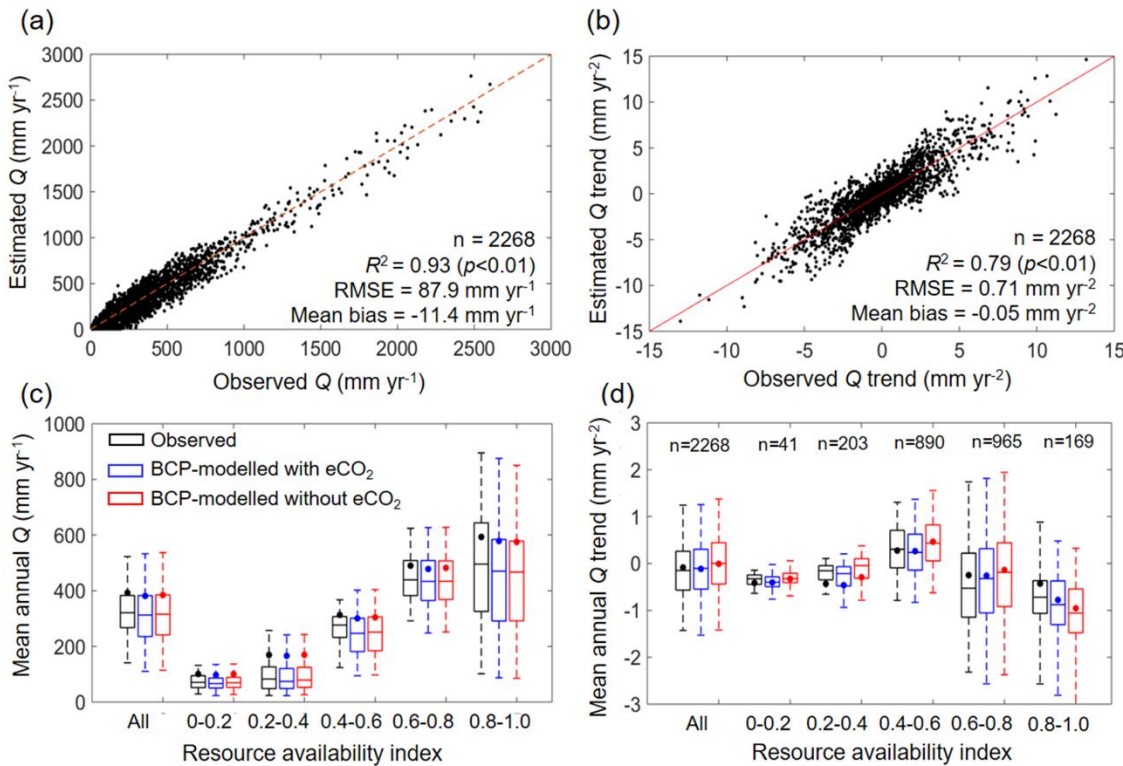

**Figure 4 Validation of estimated *Q* at catchments. a**, Model performance in predicting mean annual *Q* in 2,268 catchments over 1982-2010. **b**, Model performance in predicting *Q* trend in 2,268 catchments during 1982-2010. **c**, Model performance in predicting mean annual *Q* in 2,268 catchments over 1982-2010 stratified by resource availability index category. **d**, Model performance in predicting *Q* trend in 2,268 catchments over 1982-2010 stratified by resource availability index category. The number of catchments in each resource availability index category are provided at the top of this sub-plot. The legend from **c** applies to **d.** In **c** and **d**, the upper / lower box edges represent the quantile divisions, the inner horizontal line is the median, the dots indicate the mean, and the dashed line represent the 5% and 95% percentiles.

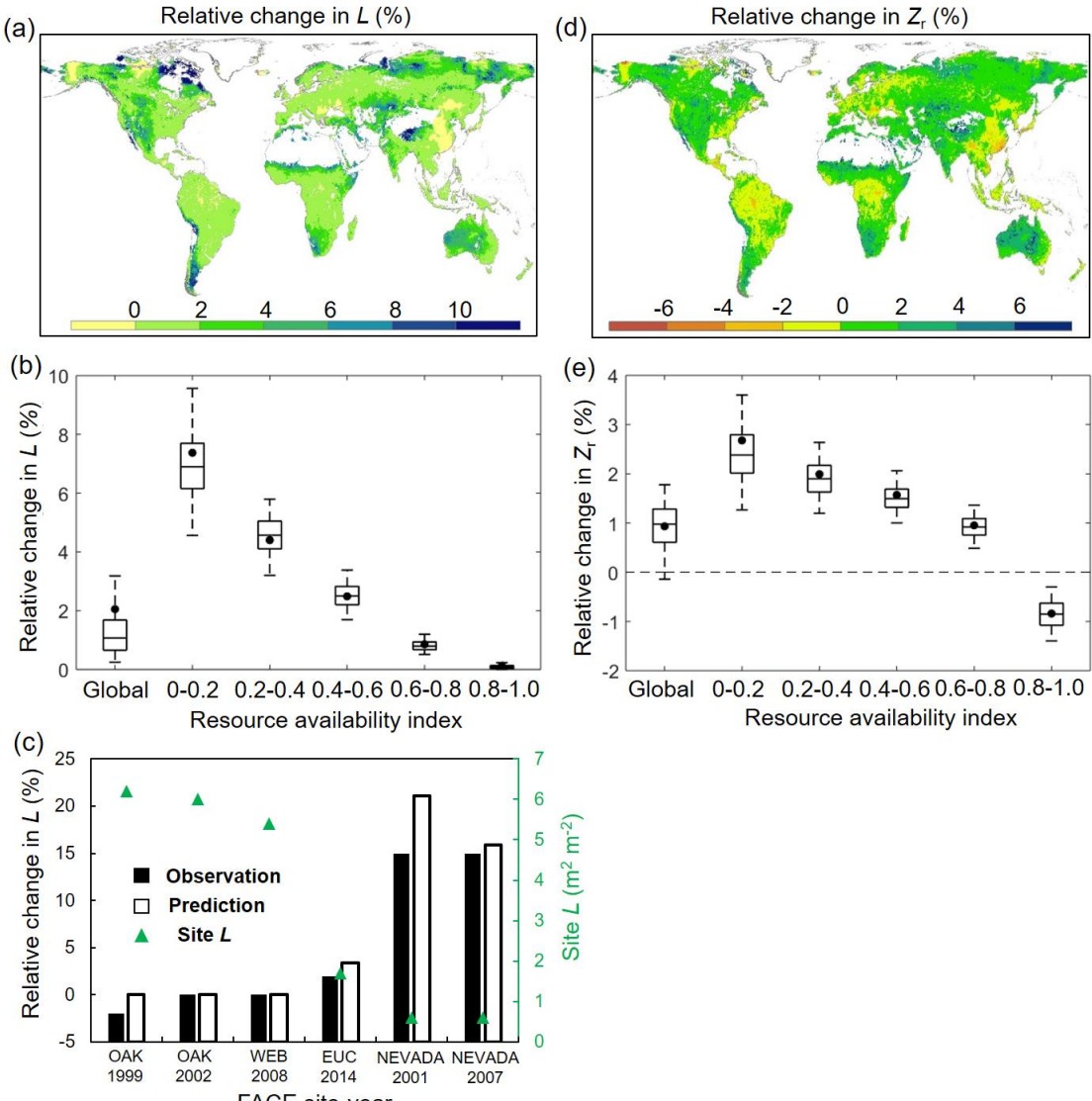

**Figure 5** Modeled relative changes in $L$ and $Z_r$ caused by eCO$_2$. **a**, Spatial distribution of relative change in $L$ induced by eCO$_2$ during 1982-2010. **b**, Same as a, but for each resource availability index category. **c**, Validation of predicted $L$ change against *in situ* measurement during six Free Air CO$_2$ Enrichment (FACE) Experiments. Note that only FACE sites with undisturbed vegetation are used (see Donohue et al., 2017). **d**, Spatial distribution of relative change in $Z_r$ induced by eCO$_2$ during 1982-2010. **e**, Same as d, but for each resource availability index category. In **b** and **e**, the upper / lower box edges represent the quantile divisions, the inner horizontal line is the median, the dots indicate the mean, and the dashed lines represent the 5% and 95% percentiles. The number of grid-cells in each resource availability index category is provided in Figure 2.

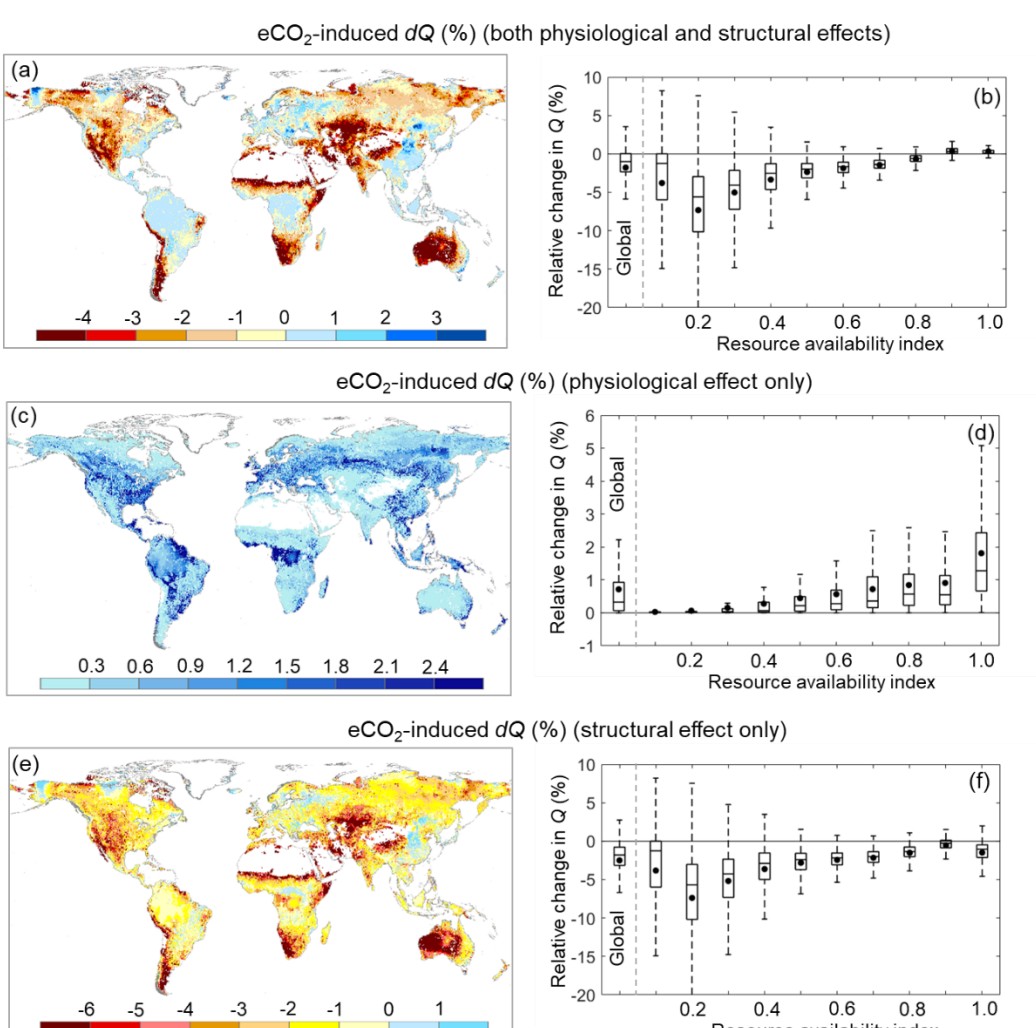

**Figure 6 Relative *Q* change induced by eCO₂ during 1982-2010 across the global vegetated lands. a**, Spatial distribution of relative change in *Q* induced by eCO₂. **b**, Boxplot of relative change in *Q* induced by eCO₂ for each resource availability index category. **c**, Spatial distribution of relative change in *Q* induced by eCO₂ when only the physiological effect is considered. **b**, Boxplot of relative change in *Q* induced by eCO₂ when only the physiological effect is considered for each resource availability index category. **e**, Spatial distribution of relative change in *Q* induced by eCO₂ when only the above-ground and below-ground structural effects are considered. **f**, Boxplot of relative change in *Q* induced by eCO₂ when only the above-ground and below-ground structural effects are considered for each resource availability index category. In **b**, **d** and **f**, the upper / lower box edges represent the quantile divisions, the inner horizontal line is the median, the dots indicate the mean, and the dashed lines represent the 5% and 95% percentiles. The number of grid-cells in each resource availability index category is provided in Figure 2.

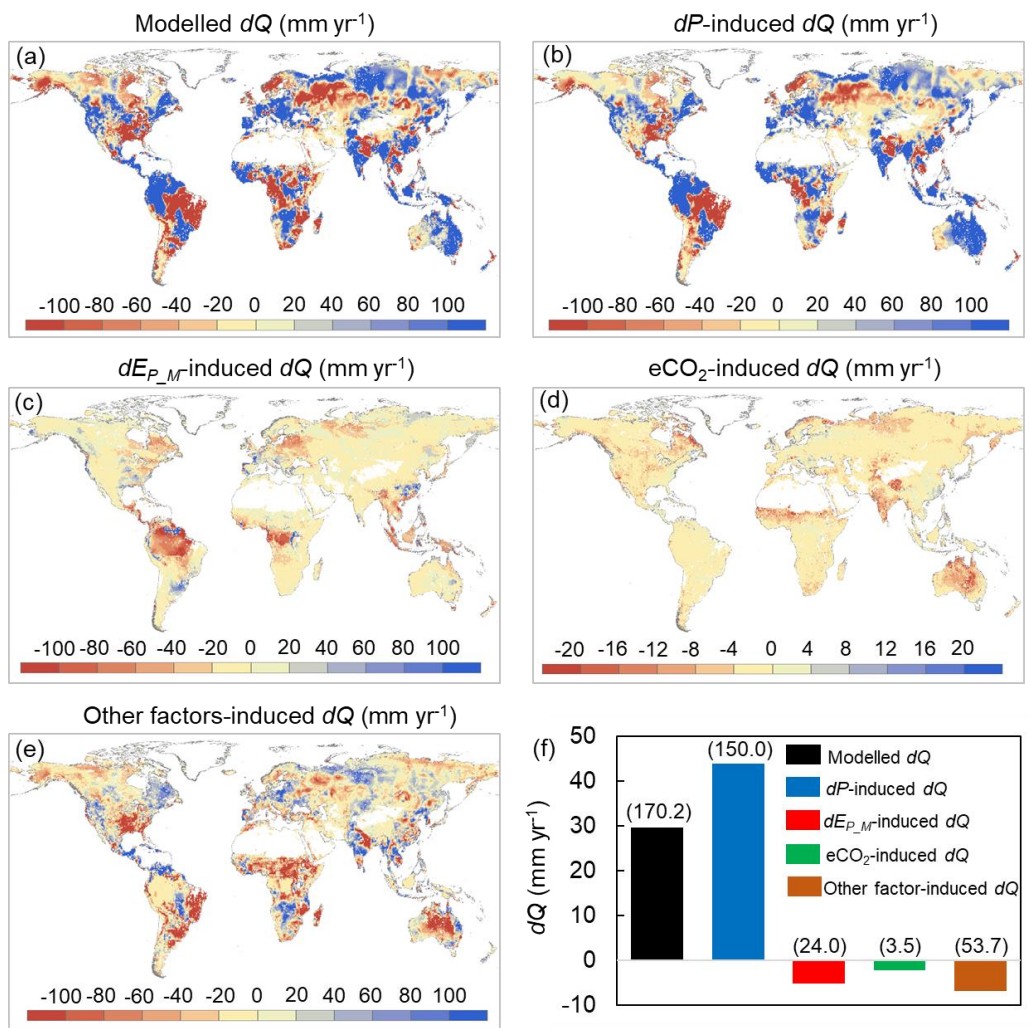

**Figure 7** Attribution of changes in $Q$ between 1982-1985 and 2006-2010 across global vegetated lands. **a**, Spatial distribution of changes in $Q$. **b-e**, Spatial distributions of changes in $Q$ induced by (b) changes in $P$, (c) changes in $E_{P\_M}$, (d) eCO$_2$, and (e) changes in other factors (mainly rainfall intensity and climate change-induced vegetation change). **f**, Attribution of changes in $Q$ between 1982-1985 and 2006-2010 averaged over the entire 805 global vegetated lands. Values in the brackets represent one standard deviation of each response among all vegetated grid-cells.

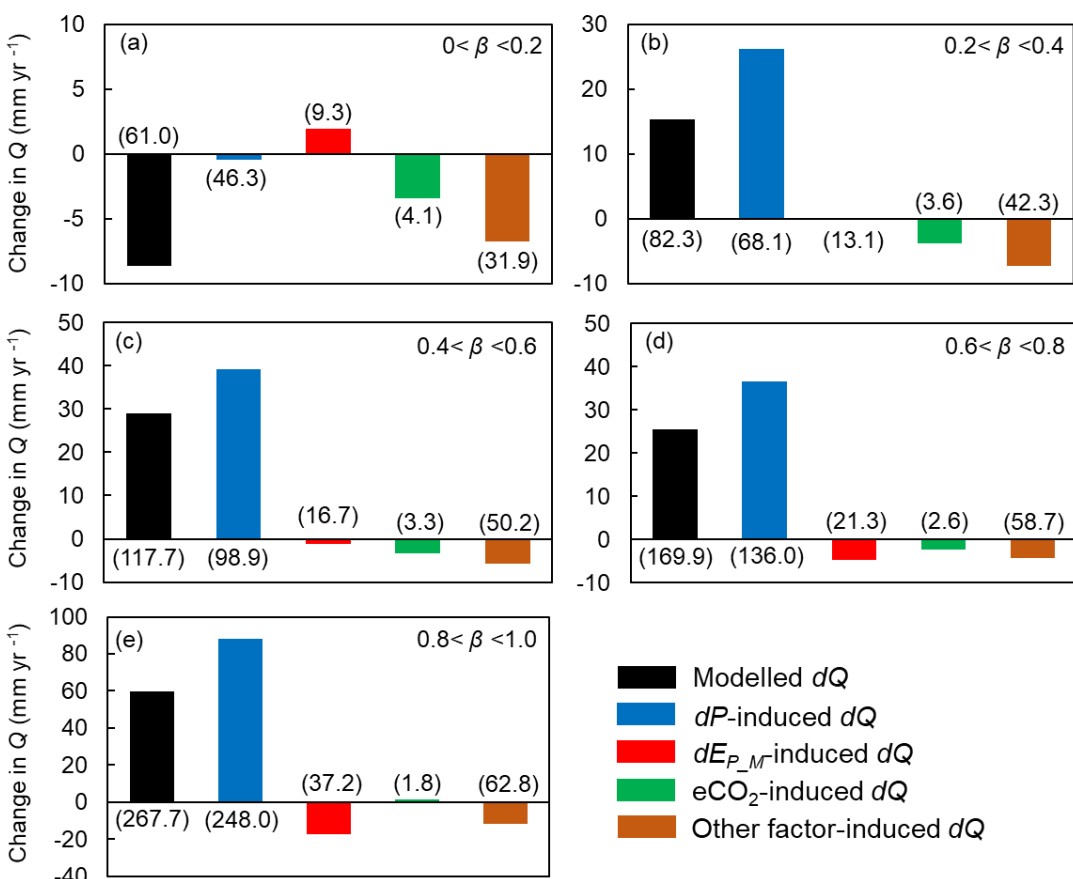

**Figure 8** Attribution of changes in $Q$ between 1982-1985 and 2006-2010 at grid-boxes within each resource availability index ($\beta$) category. Values in the brackets represent one standard deviation of each response among grid-cells within each resource availability index category. The number of grid-cells in each resource availability index category is provided in Figure 2.

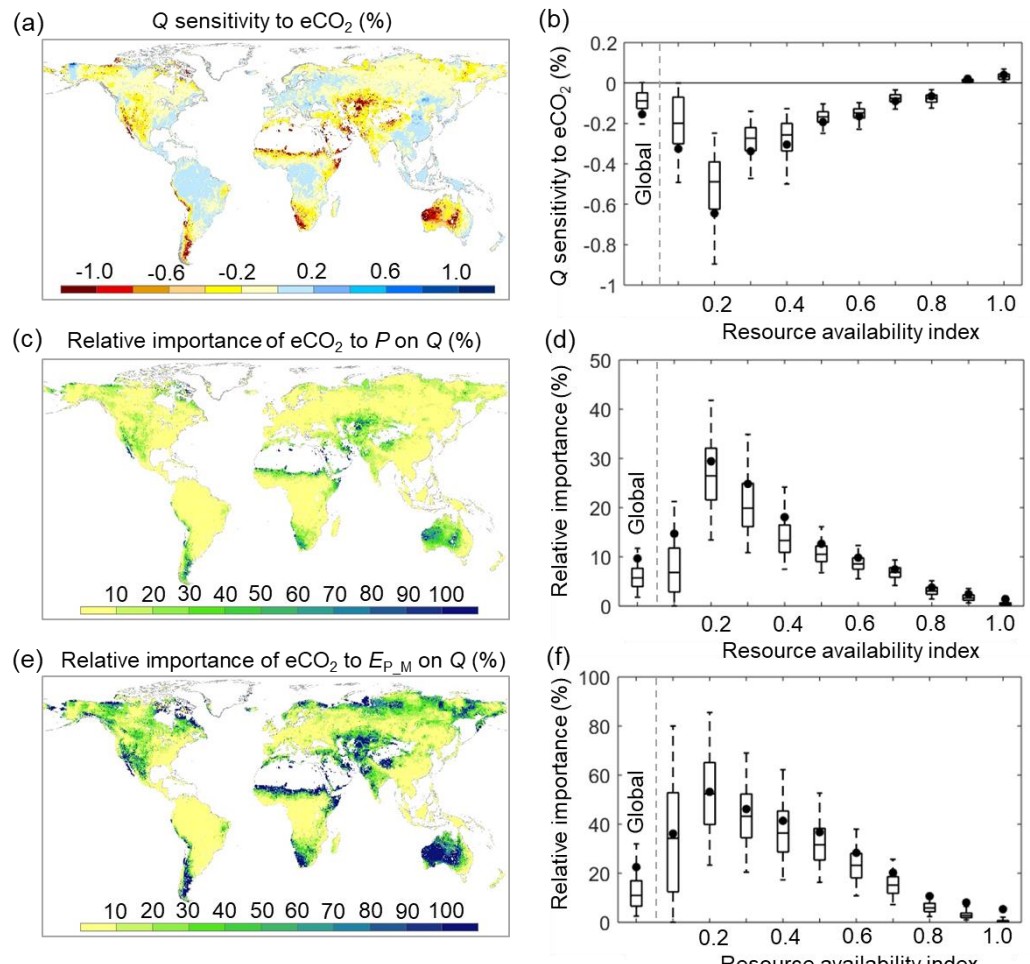

**Figure 9** Sensitivity of $Q$ to $eCO_2$ and its relative importance to $P$ and $E_{P\_M}$ across the globe. **a**, Spatial distribution of $Q$ sensitivity to $eCO_2$ (% change in $Q$ per 1% change in $C_a$). **b**, Boxplot of $Q$ sensitivity to $eCO_2$ for each resource availability index category. **c**, Relative importance of $eCO_2$ on $Q$ compared to changes in $P$ on $Q$ (% change in $Q$ per 1% change in $C_a$ compared to % change in $Q$ per 1% change in $P$). **d**, Boxplot of the relative importance of $eCO_2$ on $Q$ compared to changes in $P$ on $Q$ for each resource availability category. **e**, Relative importance of $eCO_2$ on $Q$ compared to changes in $E_{P\_M}$ on $Q$ (% change in $Q$ per 1% change in $C_a$ compared to % change in $Q$ per 1% change in $E_P$). **f**, Boxplot of the relative importance of $eCO_2$ on $Q$ compared to changes in $E_{P\_M}$ on $Q$ for each resource availability category. In **b**, **d** and **f**, the upper / lower box edges represent the quantile divisions, the inner horizontal line is the median, the dots indicate the mean value, and the dashed lines represent the 5% and 95% percentiles. The number of grid-cells in each resource availability index category is provided in Figure 2.