# Peer review of "Low and contrasting impacts of vegetation CO2 fertilization on global terrestrial runoff over 1982-2010: Accounting for above- and below-ground vegetation-CO2 effects"

_Hydrology and Earth System Sciences, 2020_

## Referee Comment (RC1) · Anonymous Referee #1 · 5 Jan 2021

This manuscript studies the effect of elevated atmospheric CO2 on runoff at the catchment scale. The approach is based on a combination of models linking elevated CO2 to plant water demand (mediated by leaf area and stomatal conductance changes) and supply (depending on soil water access via changes in rooting depth). The approach is to my knowledge novel (despite building on several previous models and data analyses) and results are interesting. The topic is certainly suitable for HESS. However, I have some concerns regarding the theoretical setup of this work, specifically how different models have been linked and the consistency of underlying modelling assumptions.

[Figure]

Other comments are also listed below.

Main concerns

- Consistency across stochastic soil water balance models. The model by Guswa (2008) assumes that actual evapotranspiration (ET) is fixed and equal to potential ET (PET) as soil moisture varies between the wilting point and saturation. In contrast, the model by Porporato et al. (2004) assumes that actual ET increases from 0 at the wilting point to PET at saturation. These two models are therefore based on different assumptions regarding the relation between actual ET and soil moisture, which in turn affect the long-term mean soil moisture and actual ET values. As a result, the ET/precipitation vs. PET/precipitation relations (i.e., relations in the Budyko space) will differ between these models. To develop a self-consistent theoretical approach to study elevated $CO_2$ effects on runoff, a single stochastic soil moisture model should be selected and used throughout. For example, see how the model by Porporato et al. (2004) can be integrated into Guswa's framework for rooting depth (Guswa, 2010, doi:10.1029/2010WR009122).

- Budyko curve parameterization. The authors use results from Porporato et al. (2004) to link the exponent n in Eq. (1) to rooting depth, water holding capacity, and mean precipitation event depth. This approach is based on analysis of "data from Porporato et al. (2004)" (L103), though it is important to emphasize that in that paper there are no data (except for net primary productivity), so the regression reported in Eq. (2) is obtained by fitting results from the analytical model in Porporato et al. (2004). This step is quite unnecessary, since the results are already in a close-form solution, which can be used directly without any fitting. In other words, Porporato et al. (2004) already provides a fully parameterized Budyko curve, which should be used for consistency with the other parts of the model instead of Eq. (1).

- Model interpretation at annual time scale. The models by both Porporato et al. (2004) and Guswa (2008) have been developed for growing season conditions, assuming no

seasonality in precipitation and potential evapotranspiration. In this contribution, these models are interpreted as representative of the whole hydrologic year and used to partition variability in annual runoff. I wonder if and how the original model assumptions and the current model interpretation can be reconciled.

- Role of precipitation event frequency. Eq. (2) neglects the effect of precipitation event frequency on the shape of the Budyko curves from Porporato et al. (2004) framework. The variations in frequency across climates can be more pronounced than variations in mean event depth.

- Interpretation of results from Donohue et al. (2013). Eq. (6) presents an iterative scheme to estimate changes in WUE through time, but in the original articles by Donohue et al. (2013, 2017) steady state models are developed, without an explicit dynamic component. The time scales to achieve steady state are probably in the order of decades (necessary for vegetation change), not years as indicated in Eq. (6).

Other comments

Notation: several symbols are defined differently from the publications they are taken from, creating some confusion. For example, mean rainfall depth is denoted by alpha (not beta) in Porporato et al. (2004); rooting depth is denoted by $Z_r$ (not $Z_e$) in Guswa (2008); symbol beta is used in Guswa (2008) as well, but has a different meaning; many symbols are used to define evapotranspiration and potential evapotranspiration, and not all are clearly defined ($E_{P_T}$, $E_T$, $E_{P_M}$, $E_P$); stomatal conductance is generally denoted by $g_s$, not $C_s$; symbol theta is used for volumetric soil moisture (not water holding capacity). To summarize, for readers familiar with the literature, reading this manuscript can be difficult because of the different meaning of commonly-used symbols.

L26: why "implicitly" - do you mean "explicitly"? L31: "the resource availability gradient" suggests that this gradient has been presented before, but it is not. L50: other recent works have discussed these issues, including Fatichi et al. (2016,

www.pnas.org/cgi/doi/10.1073/pnas.1605036113). L61: please check spelling of BCP model author names. L67 and 69: are "model parameter" and "land surface parameter" indicating the same quantity? L81: this could be a good place for a summary of the research questions or aims of the work. L96-97: just a comment - typically, ET is estimated from precipitation and runoff, since ET is the most difficult term in the catchment water balance to estimate; here the water balance is used to estimate Q, assuming the ET is known. L137: some words missing - e.g., "parameters"? L139: but evaporation from the soil surface is neglected here (L91), so I am not sure I understand this statement. L147: I would define here symbols $E_{P\_M}$ and O. L150: not clear how $E_{P\_M}$ differs from $E_P$. L156: this sentence is hard to follow. L160: singular "affects". Section 2.3: I would emphasize that this dataset covers experiments with artificially elevated CO2. L210: how was beta calculated? L236: "differentially better" - meaning not clear. L238: these statements are qualitative and no performance measure is provided to compare the two model variants. L249: "...caused an increase of L" - in the remote sensing data or based on model predictions? L252: "L increase is found..." - in the remote sensing data or based on model predictions? L265: suggested rewording "... shows a slight decrease in..." L348: I am not sure how results here can guide climate model development. Figure 3: please check units of RMSE and mean bias in panel (b). Figure 4: are the shown changes in L modelled or measured from remote sensing? Note that "but for each" in the caption is repeated. Figure 6: I suspect L587-590 are not meant to be in the caption (they seem not relevant). I would also show error bars consistent with other plots - here they represent 1/10 of standard deviation, indicating that in fact the variance is extremely large.
* * *

---

## Referee Comment (RC2) · Anonymous Referee #2 · 15 Jan 2021

The manuscript by Yang et al. aims at quantifying the impact of physiological and structural vegetation adaptations induced by elevated atmospheric CO2 concentration (eCO2) on mean annual runoff (Q). The vegetation-mediated eCO2 effect on Q is complex and involved several processes with sometimes opposite effects. Also, the link of below-ground processes to eCO2 is still not entirely clear. For these reasons, the effect of eCO2 on Q is a source of uncertainty in simulation models. This paper uses an attribution framework, based on the previously applied BCP model, to quantify the net vegetation-mediated eCO2 effect on Q. This is a highly topical subject, the choice

of methods seems appropriate and the inclusion of a link to below-ground processes constitutes a substantial novelty, which makes this manuscript of interest to HESS. However, my concerns relate to the presentation of the material: I find the manuscript difficult to follow and think that its value could be greatly increased by improving the description of methods. I therefore recommend a minor revision before the paper gets published.

I find the presentation of the methods somewhat unclear and found it difficult to understand how the different methodical steps are linked together, particularly Sections 2.3 and 2.4. Are the responses of stomatal closure and L to $eCO_2$ integrated in the BCP model? If so, please make the links explicit. If not, please clarify how these different steps work together in the attribution framework. Also, it seems to me that the step of extending the analysis from the study catchments (l. 196 states that the analysis is limited to those) to a global raster map (e.g. Fig. 7) is not described in sufficient detail in the Methods.

In the presentation of the results, it is not immediately clear if the $Q$-$eCO_2$ response refers to the net effect of increased $CO_2$ concentration on Q (through all the known effects on e.g. meteorological forcing, plant physiological and structural adaptations to $CO_2$ and climate etc – this seems to be the case in the first paragraph of Section 3.3 and Fig. 5), or the net effect of $eCO_2$-induced plant physiological and structural adaptations (this seems to be the case in the second paragraph of Section 3.3 and Fig. 6). Then again, in the first paragraph (l. 270 ff.) the authors discuss the relative importance of physiological and structural effects of $eCO_2$ on vegetation before the corresponding evidence has been presented.

The authors conclude by stating that the analyses provide insightful guidance for the development of climate models. It would be helpful to describe how exactly the findings from this analysis can be used in climate model development. In general, if this is where the value of the paper lies, it would greatly benefit from connecting the different steps (methods and discussion) to the current state of research in climate and earth system

modeling (including the significance of these feedbacks and their uncertainty for earth-system modeling, e.g. Hickler et al. 2015 https://doi.org/10.1007/s40725-015-0014-8, Li et al. 2019 https://doi.org/10.5194/bg-15-6909-2018) . For example, how does the $CO_2$ fertilization effect calculated in this paper compare to results obtained in modeling studies? How is the link between Ca and below-ground vegetation dynamics currently represented in models, and how might they benefit from the advances in this study?

Further remarks, some of them minor:

- l. 111: Please indicate the values for root respiration and the Q10 parameters.

- l. 142 ff: This is not necessarily the case. In the Guswa model, the relation of optimal rooting depth to P/EP is nonlinear and non-monotonic, with the greatest optimal depth calculated in conditions where water supply and demand are approximately equal.

- Eq. 15: please define beta.

- l.161: what exactly does "residual" mean in this context?

- Eq. 16: What are the units of S_Q_to_eCO2?

- l. 250: This average value by itself is not very informative, I suggest characterizing the distribution (mode(s) and range) in more detail (including a discussion of Fig. 4 b).

- l. 257 "has resulted": I suggest making it clearer that this statement describes simulation results, rather than observations (as I understand it).

- l.288: did you mean "other factors including"?

- l. 320: which mechanism?

- l.337: I am not sure if the word "exaggerate" corresponds to the idea expressed by the authors. Maybe "exacerbate"?

- l. 340 "This suggests that the structural response. . ." This causal link is not immediately clear to me, please clarify

- Fig. 4 a,b,d,e: To avoid any confusion I think it is important to make clear that the data shown are the results of simulations, and not (as I understand it) based on observations.

- Fig. 5: What exactly is meant by "Q change induced by eCO2" (see my comment in the 3rd paragraph)?

- Fig. 6: The size of the error bars representing 1/10 suggests a great variability of these quantities among the different catchments. Consider using an alternative visualization method (e.g. boxplots or kernel density plots).

- Fig 6: some sentence of the caption refer to elements that I cannot see (viewing the PDF in Chrome on Windows): values in parenthesis; vertical grey dashed line.
* * *

---

## Referee Comment (RC3) · Anonymous Referee #3 · 18 Jan 2021

The authors explore past runoff trends over undisturbed catchments and globally. Using an analytical framework, they attribute runoff trends to climate and vegetation influences along a resource availability index. The impact of CO2-induced vegetation changes on runoff has remained highly uncertain and as such, this study is a valuable contribution to the literature and well suited to HESS.

Whilst I find this study interesting, it is a shame it does not go further in quantifying the CO2-induced vegetation changes on Q. In particular, the authors mention the inclusion of CO2-induced rooting depth changes as a key novel aspect. However, whilst the authors quantify in detail the influence of CO2 on the individual above- and below-ground vegetation processes, it is not shown how these in turn affect Q. For Q, only the bulk CO2 response is presented if my reading of the results is correct. A number of studies already exist on the bulk CO2 responses and/or separating the effects of stomatal closure and LAI on Q (although I appreciate a new modelling framework is introduced here). Here it would have been interesting to know how Ze specifically changes Q. I think the results suggest the influence of rooting depth changes are minimal but this is glossed over in the discussion.

I would also hope more clarity on how parameter n is determined. The current explanation is not sufficient, including what data were used. The methods should also be revised for clarity, reading the results it becomes unclear what was quantified using the analytical framework vs other methods (e.g. stomatal closure and L responses). Perhaps a summary of the steps at the start of Methods would help the reader.

Specific comments:

Title: The study period doesn't cover the last three decades.

L23-26: This sentence could be written more clearly.

L30-34: This sentence should also be broken up into two for clarity.

L34: highlights -> highlight

L38: Suggest replacing "becoming" with "and representing"

L44: I would suggest Donohue is not an appropriate reference here, it is not a leaf-scale study.

L50: I think the authors need to unpack this sentence a little. Many of these studies look at the net response on Q so I'm not sure what the authors mean by "different aspects"? I would argue the main reason for the discrepancies across studies is due to the different processes and assumptions included in the models. Also Ukkola et al. is

not a modelling study but based on observations (similarly Trancoso et al. 2017 which should also be cited for an observational analysis). Some model evaluation has also been conducted specifically for CO2 impacts (e.g. Ukkola et al. 2016, Environmental Research Letters for a DGVM and multiple FACE papers), here the evaluation seems to be limited to the overall Q trends which is not new. A more accurate statement would be that observational and evaluation studies for CO2 effects remain limited, particularly at the regional to global scale.

L53, L265: please fix grammar.

L61: should be Budyko-Choudhury-Porporato

L94: I think Milly and Dunne actually found that the energy-only PET best produced non-water-stressed ET from climate models (their Figure 3, associated text and conclusions). Also climate models do not simulate potential evapotranspiration so perhaps best to avoid that terminology here?

L103: Could you be more specific here? Taking what data?

L111: Not clear to me how potential transpiration is determined?

L119: I don't see where the Earth System Models are described? Also why were ESMs used rather than something more observationally constrained? Given such a short time period is taken and coupled models have their own interannual variability, taking a mean across models over such a short time period is likely to be spurious. Why wasn't observationally-driven products used, e.g. GLEAM or the TRENDY ensemble? These are of course also models but at least driven by observed meteorology.

L138-139: Why do these quantities impact Ep? Most PET estimators are mainly atmosphere-driven so if this is not the case with Shuttleworth and Wallace, more details on its calculation need to be provided for clarity.

Equation 12: should the notation be f() instead of g()?

L218: Which years were used?

L220: You should provide the name of the dataset (i.e. ISLSCP etc.)

Figure 1: White regions in the map that do not match the colour scale (e.g. Greenland). Should say if/why these were masked out

L232, L234: missing full stop

L243: Please avoid using brackets like this, it is very hard to read. Suggest: with the largest Cs reduction found in C4 crops and lowest in shrubs.

L249: How was the Ca effect on L estimated? I'm assuming using equation 18 but it has two factors influencing L (Ca and v)

Figure 5: Would be useful to see the spatial distribution of catchment trends. Suggest adding a map of the catchments eCO2-induced trends as an additional panel

Figure 6: Last panel please adjust scale to show full error bars. Also please check caption, from L587 it mentions numbers that I don't see presented in the figure

L279: I'm confused why this result differs from the number on L269 and how the changes in Q described here differ from the previous paragraph?

L286: I'm also confused that you have suddenly moved to global results (Fig 7). In the methods, you state that the analysis is restricted to the $\sim$2000 catchments (L195). The text doesn't also make this transition obvious.

L292: Given alpha is determined from LAI, surely low-alpha regions can be either dry or cold?

L348: How exactly can this framework guide model development? Firstly, the results from this study are very much in line with existing studies so no particularly novel insights are revealed. And secondly, how is this framework to help climate model development exactly? And finally, this is ultimately simply another model result. Overall

this feels like a bit of a throw-away statement to try and boost the value of paper

L351: Are all the datasets publicly available?

---

## Author Comment (AC1) · 14 Feb 2021

**Response to Reviewers' comments**

We greatly appreciate the reviewers providing valuable and constructive comments on our manuscript HESS-2020-548. We seriously considered each comment and will revise/improve the manuscript accordingly. The individual comments are replied below. In the following the reviewer comments are black font and our responses are blue and to assist with navigation we use codes, such as R1C2 (Reviewer 1 Comment 2).

**Anonymous Referee #1**

R1C1: This manuscript studies the effect of elevated atmospheric CO2 on runoff at the catchment scale. The approach is based on a combination of models linking elevated CO2 to plant water demand (mediated by leaf area and stomatal conductance changes) and supply (depending on soil water access via changes in rooting depth). The approach is to my knowledge novel (despite building on several previous models and data analyses) and results are interesting. The topic is certainly suitable for HESS. However, I have some concerns regarding the theoretical setup of this work, specifically how different models have been linked and the consistency of underlying modelling assumptions.

Reply: Thanks for your encouraging and constructive comments. Your individual comments are replied to below.

R1C2: Consistency across stochastic soil water balance models. The model by Guswa (2008) assumes that actual evapotranspiration (ET) is fixed and equal to potential ET (PET) as soil moisture varies between the wilting point and saturation. In contrast, the model by Porporato et al. (2004) assumes that actual ET increases from 0 at the wilting point to PET at saturation. These two models are therefore based on different assumptions regarding the relation between actual ET and soil moisture, which in turn affect the long-term mean soil moisture and actual ET values. As a result, the ET/precipitation vs. PET/precipitation relations (i.e., relations in the Budyko space) will differ between these models. To develop a self-consistent theoretical approach to study elevated CO2 effects on runoff, a single stochastic soil moisture model should be selected and used throughout. For example, see how the model by Porporato et al. (2004) can be integrated into Guswa's framework for rooting depth (Guswa, 2010, doi:10.1029/2010WR009122).

Reply: We agree with this reviewer on the raised issue. In fact, we realized it when building the BCP model in 2012. The reason that we still go for Guswa-2008, instead of Guswa-2010, is that the solution of transpiration ($T$) in Porporato-2004 includes an incomplete gamma function with rooting depth contained in both parameters of that incomplete gamma function. This feature makes the analytical solution of $dT/dZ_r$ extremely complex (see below equation) and it is almost impossible to derive an explicit solution for $Z_r$. We believe this is the reason that Guswa did not provide an explicit solution of $Z_r$ in his 2010 paper. The results presented in Guswa-2010 and

Porporato-2014 were derived numerically but only for specific cases (e.g., with specified aridity index or $T_P$ or $dT/dZ_r$).

$$dT/d\gamma = -T_P*((exp(-\gamma)*(\gamma^{(W*Zr-2)}*(W*\gamma-1)+W*\gamma^{(W*\gamma-1)}*\ln(\gamma)))/(\Gamma(W*\gamma)-\Gamma(W*\gamma,\gamma))$$
$$-(\gamma^{(W*\gamma-1)}*exp(-\gamma))/(\Gamma(W*\gamma)-\Gamma(W*\gamma,\gamma))$$
$$+(\gamma^{(W*\gamma-1)}*exp(-\gamma)*(expint(1-W*\gamma,\gamma)*(W*\gamma*\gamma^{(W*\gamma-1)}+W*\gamma^{(W*\gamma)}*\ln(\gamma))$$
$$+\gamma^{(W*\gamma)}*(W*(hypergeom([W*\gamma,W*\gamma],[W*\gamma+1,W*\gamma+1],-\gamma)/(W^2*\gamma^2)$$
$$+(pi*(log(\gamma)-psi(1-W*\gamma)+pi*cot(pi*(W*\gamma-1))))/(\gamma^{(W*\gamma)}*sin(pi*(W*\gamma-1))*\Gamma(1-W*\gamma)))$$
$$-expint(-W*\gamma,\gamma))-W*\Gamma(W*\gamma)*psi(W*\gamma))/(\Gamma(W*\gamma)-\Gamma(W*\gamma,\gamma))^2)$$

and $\qquad \gamma = \dfrac{Z_r * \text{SWHC}}{\beta}$

where SWHC is soil water holding capacity and $\beta$ is the mean rainfall intensity.

In the BCP model, the Guawa's model is used to estimate the effective rooting depth, which is then used to calculate the Porporato's parameter $\omega$ (the symbol $\gamma$ is used in Porporato-2004). According to Guswa-2010, the Porporato's solution for transpiration will lead to a slightly deeper rooting depth than the Milly's solution for transpiration (adopted in Guswa-2008 and this study). Despite that, the responses of $Z_r$ to changes in climate are essentially the same when the two transpiration solutions are adopted. Moreover, the responses of $Z_r$ to changes in $CO_2$ in the two solutions should also be essentially the same, since the effects of CO2 on $Z_r$ are expressed via water use efficiency and potential transpiration in our parameterization, which are independent of $Z_r$ parameterizations. In summary, using different transpiration solutions (Milly-1993 versus Porporato-2004) would only lead to difference in the resultant absolute magnitude of runoff ($Q$) but unlikely to result in differences in the response of $Q$ to $CO_2$ changes in any notable way, especially when the relative magnitude is used.

We will discuss this point in the revised manuscript.

R1C3: Budyko curve parameterization. The authors use results from Porporato et al. (2004) to link the exponent n in Eq. (1) to rooting depth, water holding capacity, and mean precipitation event depth. This approach is based on analysis of "data from Porporato et al. (2004)" (L103), though it is important to emphasize that in that paper there are no data (except for net primary productivity), so the regression reported in Eq. (2) is obtained by fitting results from the analytical model in Porporato et al. (2004). This step is quite unnecessary, since the results are already in a close-form solution, which can be used directly without any fitting. In other words, Porporato et al. (2004) already provides a fully parameterized Budyko curve, which should be used for consistency with the other parts of the model instead of Eq. (1).
Reply: We agree with this reviewer from a theoretical perspective. However, from a practical perspective, Porporato's model is much more complex than the Budyko model and can only be solved numerically (for the reason stated in the reply to R1C2).

With a specified model parameter, Porporato et al. (2004) proved the similarity between their solution and the Budyko's solution of mean annual water balance. Compared with Porporato et al (2004), the Budyko's formulation is much simpler, which allows an analytical attribution of Q changes. Therefore, developing relationship between the Budyko's parameter (here the Choudhary's expression of the Budyko curve) and Porporato's parameter is a simple yet effective way to solve the problem. The same approach has been adopted in previous studies (e.g., Donohue et al., 2012, https://doi.org/10.1016/j.jhydrol.2012.02.033; Liu et al., 2016, https://doi.org/10.1016/j.jhydrol.2016.10.035; Yang et al., 2016, https://doi.org/10.1002/2016WR019392; Shen et al., 2017, https://doi.org/10.1016/j.jhydrol.2017.09.023; Zhang et al., 2018, https://doi.org/10.1002/2017WR022028).

This reviewer was correct that there were no data in Porporato et al. (2004). What we obtained from the authors of that paper (via email exchange in 2010) is their numerical solutions of the corresponding E/P for every 0.1 increment in PET/P for six separate $\gamma$ curves. By numerically solving the Choudhary's formulation of the Budyko curve, we determined the values of the Budyko parameter ($n$) that correspond to the E/P values of each of the six $\gamma$ curves. We then pooled all $n - \gamma$ pairs together to derive a simple relationship between them (Eq. 2 in the manuscript).

R1C4: Model interpretation at annual time scale. The models by both Porporato et al. (2004) and Guswa (2008) have been developed for growing season conditions, assuming no seasonality in precipitation and potential evapotranspiration. In this contribution, these models are interpreted as representative of the whole hydrologic year and used to partition variability in annual runoff. I wonder if and how the original model assumptions and the current model interpretation can be reconciled.
Reply: In our study, the Guswa's model was indeed applied for growing season to determine the effective rooting depth (Line xxx). The determined effective rooting depth during growing season is then used to determine the Porporato's parameter and further, the Budyko parameter. It should be noted that the effective rooting depth is essentially the maximum depth of hydrologically active soil layer, which should remain unchanged between the growing season and the whole hydrologic year.

R1C5: Role of precipitation event frequency. Eq. (2) neglects the effect of precipitation event frequency on the shape of the Budyko curves from Porporato et al. (2004) framework. The variations in frequency across climates can be more pronounced than variations in mean event depth.
Reply: We agree with this reviewer that the precipitation event frequency is important in the control of precipitation partitioning into evapotranspiration and runoff and the Porporato et al (2004)'s framework does not explicitly account for it in their model parameter. Nevertheless, the Porporato's framework considers both the total precipitation amount the mean event depth, which when combined provide information about event frequency. Therefore, the effect of variation in event

frequency on the hydrological partitioning in the Porporato's framework and the BCP model is implicitly expressed by the variations in both total precipitation amount and mean event depth.

R1C6: Interpretation of results from Donohue et al. (2013). Eq. (6) presents an iterative scheme to estimate changes in WUE through time, but in the original articles by Donohue et al. (2013, 2017) steady state models are developed, without an explicit dynamic component. The time scales to achieve steady state are probably in the order of decades (necessary for vegetation change), not years as indicated in Eq. (6).

Reply: Eq. (6) follows the gas-exchange theory at the leaf-level to quantify the response of WUE ($W$ in the following equations for simplicity) to elevated $CO_2$, originally given by Wong et al. (1979),

$$W_L = \frac{A_L}{E_{TL}} = \frac{g_s(C_a - C_i)}{1.6 g_s(v_i - v_a)} = \frac{C_a}{1.6v}(1 - \frac{C_i}{C_a})$$

where $A$ (g C m$^{-2}$ s$^{-1}$) and $E_T$ (mm s$^{-1}$) stand for the assimilation and transpiration rate, respectively, and the subscript L denotes the leaf-level variables. $C_a$ (ppm) and $C_i$ (ppm) respectively represent the ambient and intercellular concentration of $CO_2$, and $v_a$ (Pa) and $v_i$ similarly represent ambient and intercellular concentration of water vapor while $g_s$ (m s$^{-1}$) is the stomatal conductance to $CO_2$. The numeric factor 1.6 accounts for the greater diffusivity of water vapor relative to $CO_2$ in air [Wong et al., 1979]. We use $v$ to denote the leaf-to-air water vapor pressure difference (Pa), which is approximated by the atmospheric vapor pressure deficit in subsequent analysis. The relative change in $W_L$ is given by:

$$\frac{dW_L}{W_L} = \frac{dA_L}{A_L} - \frac{dE_{TL}}{E_{TL}} = \frac{dC_a}{C_a} - \frac{dv}{v} + \frac{d(1 - \frac{C_i}{C_a})}{(1 - \frac{C_i}{C_a})} \quad .$$

Observations have shown that for a given photosynthetic pathway (*i.e.*, C3 or C4 species), $C_i/C_a$ is relatively conservative [Arens et al., 2000, Long et al., 2004, Wong et al., 1979]. The response of the term $1 - C_i/C_a$ to a change in $v$ can be quantified by taking $1 - C_i/C_a$ as being approximately proportional to the square root of $v$ [Donohue et al., 2013; Farquhar et al., 1993, Medlyn et al., 2011]. Therefore, Eq. (2) can be written as:

$$\frac{dW_L}{W_L} = \frac{dA_L}{A_L} - \frac{dE_{TL}}{E_{TL}} \approx \frac{dC_a}{C_a} - \frac{1}{2}\frac{dv}{v} \quad .$$

Eq. (6) in the current manuscript is essentially the same as the above equation. This equation does not require steady-state to be satisfied. However, the above equation is for leaf-level fluxes. Applying this equation at the canopy-scale implicitly assumes the same upscaling factor when converting the leaf-level assimilation and transpiration to the canopy level for a given location. This assumption is also adopted in Donohue et al. (2013, 2017). We have made this assumption explicit in the manuscript (Line xxx).

It is also noted that Donohue et al. (2013, 3017) applied this equation at the same 5-year period as in the current study.

This reviewer did point out an important issue that this theory works better for undisturbed and mature vegetation but can be problematic for disturbed and immature vegetation (e.g., seedlings). However, the issue of vegetation age and disturbances is very complex and is well beyond the scope of this manuscript, especially considering that there are no global dataset monitoring vegetation age that we could use in our modelling. In the revised manuscript, we will point this issue out and discuss it.

R1C7: Notation: several symbols are defined differently from the publications they are taken from, creating some confusion. For example, mean rainfall depth is denoted by alpha (not beta) in Porporato et al. (2004); rooting depth is denoted by $Z_r$ (not $Z_e$) in Guswa (2008); symbol beta is used in Guswa (2008) as well, but has a different meaning; many symbols are used to define evapotranspiration and potential evapotranspiration, and not all are clearly defined ($E_{P_T}$, $E_T$, $E_{P_M}$, $E_P$); stomatal conductance is generally denoted by $g_s$, not $C_s$; symbol theta is used for volumetric soil moisture (not water holding capacity). To summarize, for readers familiar with the literature, reading this manuscript can be difficult because of the different meaning of commonly used symbols.
Reply: Thanks for this comment; we will adjust the symbols used and make use that all symbols are clearly defined in the revised manuscript.

R1C8: L26: why "implicitly" - do you mean "explicitly"?
Reply: Yes, here should be explicitly; will revise in the revised manuscript.

R1C9: L31: "the resource availability gradient" suggests that this gradient has been presented before, but it is not.
Reply: We will change it to "a resource availability gradient" in the revised manuscript.

R1C10: L50: other recent works have discussed these issues, including Fatichi et al. (2016, www.pnas.org/cgi/doi/10.1073/pnas.1605036113).
Reply: We have read the suggested paper and agree that it is very relevant and will cite it appropriately in the revised manuscript.

R1C11: L61: please check spelling of BCP model author names.
Reply: Sorry for the typo. Will fix it in the revised manuscript.

R1C12: L67 and 69: are "model parameter" and "land surface parameter" indicating the same quantity?
Reply: Yes, they are the same parameter. We will use "model parameter" throughout in the revised manuscript to avoid potential misunderstandings.

R1C13: L81: this could be a good place for a summary of the research questions or aims of the work.
Reply: Will do in the revised manuscript.

R1C14: L96-97: just a comment - typically, ET is estimated from precipitation and runoff, since ET is the most difficult term in the catchment water balance to estimate; here the water balance is used to estimate Q, assuming the ET is known.
Reply: The Choudhary's formulation of the Budyko curve expresses actual ET as a function of $P$, PET and a model parameter. Then, the assumed steady-state water balance was used to calculate $Q$ as a residual.

R1C5: L137: some words missing - e.g., "parameters"?
Reply: Oops! Will fix it in the revised manuscript.

R1C16: L139: but evaporation from the soil surface is neglected here (L91), so I am not sure I understand this statement.
Reply: We believe this reviewer misunderstood our approach. L91 reads "by taking soil surface resistance equal to zero". This mean that soil evaporation would occur at its potential rate.

R1C17: L147: I would define here symbols E_{P_M} and O.
Reply: We will define the symbol as suggested in the revised manuscript.

R1C18: L150: not clear how E_{P_M} differs from E_P.
Reply: $E_{P\_M}$ is the potential evapotranspiration that is only affected by meteorological conditions, while $E_P$ is the potential evapotranspiration that is affected by both meteorological factors and CO2 concentration. With the increase of CO2, stomatal conductance decreases and LAI increases, both of which will affect potential evapotranspiration (actual evapotranspiration rate when water is not limiting). We will make the definition of $E_P$ and $E_{P\_M}$ clear in the revised manuscript.

R1C19: L156: this sentence is hard to follow.
Reply: We will rephrase this sentence in the revised manuscript.

R1C20: L160: singular "affects".
Reply: Will do as suggested; thanks.

R1C21: Section 2.3: I would emphasize that this dataset covers experiments with artificially elevated CO2.
Reply: Will do as suggested; thanks.

R1C22: L210: how was beta calculated?
Reply: We will make this clear in the revised manuscript.

R1C23: L236: "differentially better" - meaning not clear.
Reply: We will rephrase it in the revised manuscript.

R1C24: L238: these statements are qualitative and no performance measure is provided to compare the two model variants.
Reply: We will add statistics to support this statement in the revised manuscript.

R1C25: L249: ". . .caused an increase of L" - in the remote sensing data or based on model predictions?
Reply: This is model prediction. The increased L reported here is only driven by the fertilization effect. We will explicitly state that this is a model prediction in the revised manuscript.

R1C26: L252: "L increase is found. . ." - in the remote sensing data or based on model predictions?
Reply: This is model prediction too. Please see our reply to R1C25.

R1C27: L265: suggested rewording ". . . shows a slight decrease in. . ."
Reply: Will do as suggested; thanks.

R1C28: L348: I am not sure how results here can guide climate model development.
Reply: We will reword this part in the revised manuscript.

R1C29: Figure 3: please check units of RMSE and mean bias in panel (b).
Reply: Oops! We will correct it in the revised manuscript.

R1C29: Figure 4: are the shown changes in L modelled or measured from remote sensing? Note that "but for each" in the caption is repeated.
Reply: This is modelled LAI change driven by the fertilization effect. Please see our reply to R1C25.

R1C30: Figure 6: I suspect L587-590 are not meant to be in the caption (they seem not relevant). I would also show error bars consistent with other plots – here they represent 1/10 of standard deviation, indicating that in fact the variance is extremely large.
Reply: We will correct the issues in the caption; please accept our apologies. The reviewer is correct that the variance among catchments is very large. This is because runoff and its changes differs quite a lot among catchments (Figures 3). However, it can be seen from Figure 6 that the variances are larger for climate change-induced runoff changes (precipitation and potential evapotranspiration). This is also reasonable, because the variances of P and PET changes can also be large among catchments. In direct contrast, the eCO2-induced runoff changes show a much smaller variance. We showed "1/10 of standard deviation" to that to better illustrate differences in means.

Thanks for your constructively critical review / assessment of our manuscript which provided the catalyst for improvement.

**Anonymous Referee #2**

R2C1: The manuscript by Yang et al. aims at quantifying the impact of physiological and structural vegetation adaptations induced by elevated atmospheric $CO_2$ concentration (eCO2) on mean annual runoff (Q). The vegetation-mediated eCO2 effect on Q is complex and involved several processes with sometimes opposite effects. Also, the link of below-ground processes to eCO2 is still not entirely clear. For these reasons, the effect of eCO2 on Q is a source of uncertainty in simulation models. This paper uses an attribution framework, based on the previously applied BCP model, to quantify the net vegetation-mediated eCO2 effect on Q. This is a highly topical subject, the choice of methods seems appropriate and the inclusion of a link to below-ground processes constitutes a substantial novelty, which makes this manuscript of interest to HESS. However, my concerns relate to the presentation of the material: I find the manuscript difficult to follow and think that its value could be greatly increased by improving the description of methods. I therefore recommend a minor revision before the paper gets published.
Reply: Thanks for your favorable evaluation of our study. Your individual comments are replied to below.

R2C2: I find the presentation of the methods somewhat unclear and found it difficult to understand how the different methodical steps are linked together, particularly Sections 2.3 and 2.4. Are the responses of stomatal closure and L to eCO2 integrated in the BCP model? If so, please make the links explicit. If not, please clarify how these different steps work together in the attribution framework. Also, it seems to me that the step of extending the analysis from the study catchments (l. 196 states that the analysis is limited to those) to a global raster map (e.g. Fig. 7) is not described in sufficient detail in the Methods.
Reply: Thanks for your suggestion. We will document the methods more clearly in the revised manuscript.

R2C3: In the presentation of the results, it is not immediately clear if the Q-eCO2 response refers to the net effect of increased $CO_2$ concentration on Q (through all the known effects on e.g. meteorological forcing, plant physiological and structural adaptations to CO2 and climate etc – this seems to be the case in the first paragraph of Section 3.3 and Fig. 5), or the net effect of eCO2-induced plant physiological and structural adaptations (this seems to be the case in the second paragraph of Section 3.3 and Fig. 6). Then again, in the first paragraph (l. 270 ff.) the authors discuss the relative importance of physiological and structural effects of eCO2 on vegetation before the corresponding evidence has been presented.

Reply: The reported effect does not include the CO2 effect on meteorological forcing but only for the CO2 effect on Q via vegetation feedbacks (plant physiological and structure responses). Thanks for your comment and we will make this clear in the revised manuscript.

R2C4: The authors conclude by stating that the analyses provide insightful guidance for the development of climate models. It would be helpful to describe how exactly the findings from this analysis can be used in climate model development. In general, if this is where the value of the paper lies, it would greatly benefit from connecting the different steps (methods and discussion) to the current state of research in climate and earth system modeling (including the significance of these feedbacks and their uncertainty for earth system modeling, e.g. Hickler et al. 2015 https://doi.org/10.1007/s40725-015-0014-8, Li et al. 2019 https://doi.org/10.5194/bg-15-6909-2018) . For example, how does the CO2 fertilization effect calculated in this paper compare to results obtained in modeling studies? How is the link between Ca and below-ground vegetation dynamics currently represented in models, and how might they benefit from the advances in this study?
Reply: Thanks for your suggestion. We will reword this part in the revised manuscript

R2C5: l. 111: Please indicate the values for root respiration and the Q10 parameters.
Reply: Will do as suggested; thanks.

R2C6: l. 142 ff: This is not necessarily the case. In the Guswa model, the relation of optimal rooting depth to P/EP is nonlinear and non-monotonic, with the greatest optimal depth calculated in conditions where water supply and demand are approximately equal.
Reply: The statement here discusses the potential $CO_2$ fertilization effect on rooting depth and not how rooting depth would respond to climate variations. To avoid confusion, we will make this explicit in our revision.

R2C7: Eq. 15: please define beta.
Reply: Beta was firstly introduced in line 101 and used in Eq. 2.

R2C8: l.161: what exactly does "residual" mean in this context?
Reply: Residual here refers to total $dQ$ minus $P$-induced $dQ$ plus $E_{P\_M}$-induced $dQ$ plus $CO_2$-induced $dQ$. In our parameterization, this other effect may include $dQ$ caused by changes in rainfall intensity and climate-driven vegetation changes. We will make this clear in the revised manuscript.

R2C9: Eq. 16: What are the units of S_Q_to_eCO2?
Reply: Apologies for our oversight; the units of $S_{Q\_to\_CO2}$ in Eq. 16 is mm yr$^{-1}$ ppm$^{-1}$. We will add the units in the revised manuscript.

R2C10: l. 250: This average value by itself is not very informative, I suggest

characterizing the distribution (mode(s) and range) in more detail (including a discussion of Fig. 4 b).
Reply: Will do as suggested; thanks.

R2C11: l. 257 "has resulted": I suggest making it clearer that this statement describes simulation results, rather than observations (as I understand it).
Reply: Good point and yes we will do as suggested as trends from simulation have less 'scientific weight' than observed trends; thanks. We will say something like "the simulated resulted showed" or something similar.

R2C12: l.288: did you mean "other factors including"?
Reply: Oops! Will revise as suggested.

R2C13: l. 320: which mechanism?
Reply: We will reword this in the revised manuscript to avoid it being potentially ambiguous.

R2C14: l.337: I am not sure if the word "exaggerate" corresponds to the idea expressed by the authors. Maybe "exacerbate"?
Reply: Sorry for the typo. Will be revise as suggested. Thanks for your careful review; its appreciated.

R2C15: l. 340 "This suggests that the structural response. . ." This causal link is not immediately clear to me, please clarify
Reply: We will reword this part to make it clear in the revised manuscript. Basically, the structural response of vegetation to $eCO_2$ decreases with the increase of leaf area index, so with the increases of leaf area index in future climate scenarios (due to higher $P$ and $CO_2$), the response of vegetation structure (e.g., leaf area index) to elevated $CO_2$ will eventually decreases.

R2C16: Fig. 4 a,b,d,e: To avoid any confusion I think it is important to make clear that the data shown are the results of simulations, and not (as I understand it) based on observations.
Reply: Thanks for the suggestion. Will do as suggested, and please see our reply to R2C11 above.

R2C17: Fig. 5: What exactly is meant by "Q change induced by eCO2" (see my comment in the 3rd paragraph)?
Reply: We will make it clear that this $eCO_2$ effect only refers to the $eCO_2$-induced effects via vegetation feedbacks on $Q$ in the revised manuscript.

R2C18: Fig. 6: The size of the error bars representing 1/10 suggests a great variability of these quantities among the different catchments. Consider using an alternative visualization method (e.g. boxplots or kernel density plots).

Reply: Good point; we will find a better way to present this result in the revised manuscript. Please see our reply to R1C30.

R2C19: Fig 6: some sentence of the caption refer to elements that I cannot see (viewing the PDF in Chrome on Windows): values in parenthesis; vertical grey dashed line.
Reply: We will correct this issue in the revised manuscript.

Thanks very much for your careful and detailed review which will result in our manuscript improving.

**Anonymous Referee #3**

R3C1: The authors explore past runoff trends over undisturbed catchments and globally. Using an analytical framework, they attribute runoff trends to climate and vegetation influences along a resource availability index. The impact of CO2-induced vegetation changes on runoff has remained highly uncertain and as such, this study is a valuable contribution to the literature and well suited to HESS.
Reply: Thanks for your encouraging and constructive comments. We reply to your individual comments below.

R3C2: Whilst I find this study interesting, it is a shame it does not go further in quantifying the CO2-induced vegetation changes on Q. In particular, the authors mention the inclusion of CO2-induced rooting depth changes as a key novel aspect. However, whilst the authors quantify in detail the influence of CO2 on the individual above- and below-ground vegetation processes, it is not shown how these in turn affect Q. For Q, only the bulk CO2 response is presented if my reading of the results is correct. A number of studies already exist on the bulk CO2 responses and/or separating the effects of stomatal closure and LAI on Q (although I appreciate a new modelling framework is introduced here). Here it would have been interesting to know how Ze specifically changes Q. I think the results suggest the influence of rooting depth changes are minimal but this is glossed over in the discussion.
Reply: This is a very good suggestion; thanks. However, in our framework, the impacts of $eCO_2$ on vegetation structures (both LAI and rooting depth) are effectively calculated simultaneously, making it difficult to examine the LAI-induced and rooting depth-induced runoff changes separately. Specifically, $eCO_2$-induced rooting depth change is parameterized through changes in WUE and potential transpiration, and the $eCO_2$-induced potential transpiration change is caused by $eCO_2$-induced changes in stomatal conductance and LAI. Hence, the framework developed here does not allow a separate evaluation of LAI and rooting depth effects. Following your suggestion, we will perform new analysis to examine how the physiological and structural responses of vegetation to $eCO_2$ affect runoff separately.

R3C3: I would also hope more clarity on how parameter n is determined. The current explanation is not sufficient, including what data were used. The methods should also be revised for clarity, reading the results it becomes unclear what was quantified using the analytical framework vs other methods (e.g. stomatal closure and L responses). Perhaps a summary of the steps at the start of Methods would help the reader.
Reply: Thanks for the suggestion. We will further clarify the methods in the revised manuscript. Please see our response to R2C2, above.

R3C4: Title: The study period doesn't cover the last three decades.
Reply: to be explicit regarding the study period in the title will change it to be "Low and contrasting impacts of vegetation $CO_2$ fertilization on global terrestrial runoff over 1982-2010: Accounting for above- and below-ground vegetation-$CO_2$ effects"

R3C5: L23-26: This sentence could be written more clearly.
Reply: We will rephrase this sentence to make it more clear.

R3C6: L30-34: This sentence should also be broken up into two for clarity.
Reply: Will do as suggested; thanks.

R3C7: L34: highlights -> highlight
Reply: Will do as suggested; thanks.

R3C8: L38: Suggest replacing "becoming" with "and representing"
Reply: Will do as suggested; thanks.

R3C9: L44: I would suggest Donohue is not an appropriate reference here, it is not a leaf-scale study.
Reply: We will delete this reference and insert an appropriate leaf-scale study citation here.

R3C10: L50: I think the authors need to unpack this sentence a little. Many of these studies look at the net response on Q so I'm not sure what the authors mean by "different aspects"? I would argue the main reason for the discrepancies across studies is due to the different processes and assumptions included in the models. Also Ukkola et al. is not a modelling study but based on observations (similarly Trancoso et al. 2017 which should also be cited for an observational analysis). Some model evaluation has also been conducted specifically for CO2 impacts (e.g. Ukkola et al. 2016, Environmental Research Letters for a DGVM and multiple FACE papers), here the evaluation seems to be limited to the overall Q trends which is not new. A more accurate statement would be that observational and evaluation studies for CO2 effects remain limited, particularly at the regional to global scale.
Reply: Thanks for the suggestion. We will rephrase this part in the revised manuscript as suggested.

R3C11: L53, L265: please fix grammar.
Reply: Will do; thanks.

R3C12: L61: should be Budyko-Choudhury-Porporato
Reply: Oops! Will revised as suggested; thanks.

R3C13: L94: I think Milly and Dunne actually found that the energy-only PET best produced non-water-stressed ET from climate models (their Figure 3, associated text and conclusions). Also climate models do not simulate potential evapotranspiration so perhaps best to avoid that terminology here?
Reply: Milly and Dunne (2016) used the two-source Shuttleworth-Wallace model to examine the effect of $eCO_2$ on vegetation stomatal conductance and the consequent impact on potential evapotranspiration (the actual evapotranspiration when water is not limiting) (their Figure 2 and Figure 3). The calculation of PET using the two-source Shuttleworth-Wallace model in our study follows Milly and Dunne (2016; https://doi.org/10.1038/nclimate3046). Yang et al. (2019; https://doi.org/10.1038/s41558-018-0361-0) further demonstrated that why Milly and Dunne found energy-only PET works well is because that the warming-induced PET increases are almost entirely offset by the $eCO_2$-induced PET decreases. By using the two-source Shuttleworth-Wallace model, we are able to explicitly consider the impacts of LAI and conductance changes on PET.

We agree that the climate models do not simulate PET. We will rephrase this statement in the revised manuscript.

R3C14: L103: Could you be more specific here? Taking what data?
Reply: The data obtained from the authors of that paper is their numerical solutions of the corresponding E/P for every 0.1 increment in PET/P for six separate $\gamma$ curves. By numerically solving the Choudhary's formulation of the Budyko curve, we determined the values of the Budyko parameter ($n$) that correspond to the E/P values of each of the six $\gamma$ curves. We then pooled all $n - \gamma$ pairs together to derived a simple relationship between them (Eq. 2 in the manuscript).

We will make this clear in the revised manuscript.

R3C15: L111: Not clear to me how potential transpiration is determined?
Reply: Potential transpiration was determined by applying the two-source Shuttleworth-Wallace model while assuming a non-water-limiting condition. This two-source approach allows evaporation from soil and transpiration from vegetation to be calculated separately. We will extend the description of potential evapotranspiration calculations to make it clear in the revised manuscript.

R3C16: L119: I don't see where the Earth System Models are described? Also why were ESMs used rather than something more observationally constrained? Given such

a short time period is taken and coupled models have their own interannual variability, taking a mean across models over such a short time period is likely to be spurious. Why wasn't observationally-driven products used, e.g. GLEAM or the TRENDY ensemble? These are of course also models but at least driven by observed meteorology

Reply: We are sorry that we missed the description of the ecosystem models used in this study. To obtain a spatial pattern of *WUE*, global monthly GPP and $E_T$ estimates over 1982-1985 were obtained from 8 ecosystem models from MsTMIP (Multi-scale Synthesis and Terrestrial Model Intercomparison Project; *Huntzinger et al.* [2013]), including: (i) CLM [*Mao et al.*, 2012]; (ii) CLM4-VIC [*Li et al.*, 2011]; (iii) ISAM [*Jain et al.*, 1996]; (iv) TRIPLEX [*Peng et al.*, 2002]; (v) LPJ-wsl [*Sitch et al.*, 2003]; (vi) ORCHIDEE-LSCE [*Krinner et al.*, 2005]; (vii) SiBCASA [*Schaefer et al.*, 2008]; and (viii) VISIT [*Ito*, 2010]. MsTMIP is a model comparison program, which is similar to TRENDY. All models participated in MsTMIP are forced by observed meteorological variables. In fact, the same modelling outputs of these ecosystem models contributed to MsTMIP and TRENDY at the same time. We will add relevant descriptions in the revised manuscript.

R3C17: L138-139: Why do these quantities impact Ep? Most PET estimators are mainly atmosphere-driven so if this is not the case with Shuttleworth and Wallace, more details on its calculation need to be provided for clarity.

Reply: The Shuttleworth-Wallace model is a two-source evapotranspiration model, which calculates soil evaporation and plant transpiration separately. To be able to distinguish the two flux sources, vegetation conditions are needed in the model. We will extend the descriptions of the Shuttleworth-Wallace model and how potential evapotranspiration and its components (potential evaporation and potential transpiration) are calculated in the revised manuscript.

R3C18: Equation 12: should the notation be f() instead of g()?

Reply: Using f() and g() indicate that the two functional relationships are different.

R3C19: L218: Which years were used?

Reply: The year of 2001 was used. We will add this information in the revised manuscript.

R3C20: L220: You should provide the name of the dataset (i.e. ISLSCP etc.)

Reply: Will do as suggested; thanks.

R3C21: Figure 1: White regions in the map that do not match the colour scale (e.g. Greenland). Should say if/why these were masked out

Reply: Will do as suggested; thanks.

R3C22: L232, L234: missing full stop

Reply: Oops! Will correct it in the revised manuscript.

R3C23: L243: Please avoid using brackets like this, it is very hard to read. Suggest: with the largest Cs reduction found in C4 crops and lowest in shrubs.
Reply: Will revise as suggested; thanks.

R3C24: L249: How was the Ca effect on L estimated? I'm assuming using equation 18 but it has two factors influencing L (Ca and v)
Reply: This reviewer was correct that both Ca and $v$ are considered in our estimation of how $L$ respond to changes in Ca. The reason that L increases with elevated $CO_2$ is because that WUE increases with elevated $CO_2$. However, the response of WUE to elevated $CO_2$ is also mediated by changes in $v$. In our modelling framework, we firstly estimate how WUE changes and then how $L$ respond to changes in WUE. In the revised manuscript, we will try to rephrase relevant text to avoid potential misunderstandings.

R3C25: Figure 5: Would be useful to see the spatial distribution of catchment trends. Suggest adding a map of the catchments eCO2-induced trends as an additional panel
Reply: Will do as suggested; thanks.

R3C26: Figure 6: Last panel please adjust scale to show full error bars. Also please check caption, from L587 it mentions numbers that I don't see presented in the figure
Reply: Will do as suggested; thanks.

R3C27: L279: I'm confused why this result differs from the number on L269 and how the changes in Q described here differ from the previous paragraph?
Reply: Oops! Sorry for the typo. The change should be -2.3 mm yr$^{-1}$. We will correct this number on L269 in the revised manuscript.

R3C28: L286: I'm also confused that you have suddenly moved to global results (Fig 7). In the methods, you state that the analysis is restricted to the ~2000 catchments (L195). The text doesn't also make this transition obvious.
Reply: Sorry about that, we will add necessary information in the Methods section and revise relevant text in the Results section to make this transition more smoothly.

R3C29: L292: Given alpha is determined from LAI, surely low-alpha regions can be either dry or cold?
Reply: Yes, this reviewer was correct.

R3C30: L348: How exactly can this framework guide model development? Firstly, the results from this study are very much in line with existing studies so no particularly novel insights are revealed. And secondly, how is this framework to help climate model development exactly? And finally, this is ultimately simply another model result. Overall this feels like a bit of a throw-away statement to try and boost the value of paper.

Reply: We will reword this part to avoid overstatement.

R3C31: L351: Are all the datasets publicly available?
Reply: Yes, they are publicly available. We will make sure that all data sources are explicitly documented in the revised manuscript.

Finally, thanks very much for your review which has allowed us to improve the science in our manuscript.

---

## Author Response (AR1)

**Response to Reviewers' comments**

We greatly appreciate the reviewers providing valuable and constructive comments on our manuscript HESS-2020-548. We seriously considered each comment and revised/improved the manuscript accordingly. The individual comments are replied below.

**Editor:**

Three reviewers have given feedback to your manuscript. They all acknowledge the significance of the topic and value for the readership of HESS, and I agree with them. They all also give constructive comments on how the manuscript can be improved in terms of the explanation of the methods and additional analysis. You have already responded to the comments in your final comments to the discussion. If implemented as suggested, I believe they would address most of the concerns raised. Improving the methods section and adding analysis that allows evaluating the role of the rooting depth will definitely enhance the impact of the manuscript. Some of the responses to reviewer 1, were not explicitly announced to become part of the new version of the manuscript (e.g. R1C3, R1C4, R1C5). I believe your answers would provide valuable information for evaluating the method to the readership beyond the review process, and I strongly encourage you to incorporate those.

Reply: Thanks for your encouraging comments and the opportunity to revise the manuscript. We have seriously considered individual comments with the main changes in the new version include:

(1) Following R2C2, R3C3, R3C14, R3C15 and R3C17, we added a beginning paragraph in the Method section and Figure 1, which summarize our modelling approach and how the physiological and structural responses of vegetation to $eCO_2$ are considered in our modelling framework to detect the impact of $eCO_2$ on $Q$. The method section is also extended to include all details of our calculation procedures.

(2) Following R2C2 and R3C28, we removed the catchment scale attribution analysis (i.e., what was Fig 5 in our original submission) and along the lines their comments applied the analysis across the entire global vegetated lands with Figures 6 to 8 being new analysis augmenting the Figures 5 to 6 of our original submission. The catchments $Q$ observations are only used to validate the modelled $Q$ (now Figure 4 being Figure 3 in our original submission).

(3) Following R1C2, R1C3 and your suggestion, we added a paragraph discussing the potential limitations of our modelling framework (Line 418 to 453).

By implementing your and the reviewers comments our manuscript has greatly improved when compared to our original submission. We look forward to receiving any further comments from you and the reviewers; thank you.

Please see our full response below.
To help editor and reviewers, the following colour scheme is used in this response letter:

- Original reviewer comments in **black**
- Our response in **blue**
- New or revised manuscript text in **green**.

To assist with navigation and cross-referencing between reviewers we use codes for each comment. With R1C1 meaning 'Reviewer 1 Comment 1' and so on.

**Anonymous Referee #1**

(Black = original reviewer comments; blue = our response; and green = new or revised text).

R1C1: This manuscript studies the effect of elevated atmospheric CO2 on runoff at the catchment scale. The approach is based on a combination of models linking elevated CO2 to plant water demand (mediated by leaf area and stomatal conductance changes) and supply (depending on soil water access via changes in rooting depth). The approach is to my knowledge novel (despite building on several previous models and data analyses) and results are interesting. The topic is certainly suitable for HESS. However, I have some concerns regarding the theoretical setup of this work, specifically how different models have been linked and the consistency of underlying modelling assumptions.

Reply: Thanks for your encouraging and constructive comments. Your individual comments are replied to below.

R1C2: Consistency across stochastic soil water balance models. The model by Guswa (2008) assumes that actual evapotranspiration (ET) is fixed and equal to potential ET (PET) as soil moisture varies between the wilting point and saturation. In contrast, the model by Porporato et al. (2004) assumes that actual ET increases from 0 at the wilting point to PET at saturation. These two models are therefore based on different assumptions regarding the relation between actual ET and soil moisture, which in turn affect the long-term mean soil moisture and actual ET values. As a result, the ET/precipitation vs. PET/precipitation relations (i.e., relations in the Budyko space) will differ between these models. To develop a self-consistent theoretical approach to study elevated CO2 effects on runoff, a single stochastic soil moisture model should be selected and used throughout. For example, see how the model by Porporato et al. (2004) can be integrated into Guswa's framework for rooting depth (Guswa, 2010, doi:10.1029/2010WR009122).

Reply: We agree and realized this issue when building the BCP model in 2012. The reason that we still use Guswa-2008, instead of Guswa-2010, is that the solution of transpiration ($T$) in Porporato-2004 includes an incomplete gamma function with rooting depth contained in both parameters of that incomplete gamma function. This feature makes the analytical solution of d$T$/d$Z_r$ extremely complex (see the equation below) and it is almost impossible to derive an explicit solution for $Z_r$. We believe this is the reason that Guswa did not provide an explicit solution of $Z_r$ in his 2010 paper. The results presented in Guswa-2010 and Porporato-2014 were derived numerically but only for specific cases (e.g., with specified aridity index or $T_P$ or d$T$/d$Z_r$).

$$dT / d\gamma = -T_P * ((exp(-\gamma) * (\gamma^{(W*Zr-2)} * (W * \gamma - 1) + W * \gamma^{(W*\gamma-1)} * \ln(\gamma))) / (\Gamma(W * \gamma) - \Gamma(W * \gamma, \gamma))$$
$$-(\gamma^{(W*\gamma-1)} * exp(-\gamma)) / (\Gamma(W * \gamma) - \Gamma(W * \gamma, \gamma))$$
$$+(\gamma^{(W*\gamma-1)} * exp(-\gamma) * (expint(1 - W * \gamma, \gamma) * (W * \gamma * \gamma^{(W*\gamma-1)} + W * \gamma^{(W*\gamma)} * \ln(\gamma))$$
$$+\gamma^{(W*\gamma)} * (W * (hypergeom([W * \gamma, W * \gamma], [W * \gamma + 1, W * \gamma + 1], -\gamma) / (W^2 * \gamma^2)$$
$$+(pi * (log(\gamma) - psi(1 - W * \gamma) + pi * cot(pi * (W * \gamma - 1)))) / (\gamma^{(W*\gamma)} * sin(pi * (W * \gamma - 1)) * \Gamma(1 - W * \gamma)))$$
$$-expint(-W * \gamma, \gamma)) - W * \Gamma(W * \gamma) * psi(W * \gamma))) / (\Gamma(W * \gamma) - \Gamma(W * \gamma, \gamma))^2)$$

and $\qquad \gamma = \dfrac{Z_r * \text{WHC}}{\alpha}$

where WHC is soil water holding capacity and $\alpha$ is the mean rainfall intensity.

In the BCP model, the Guswa's (2008) model is used to estimate the effective rooting depth, which is then used to calculate the Porporato's parameter $\omega$ (the symbol $\gamma$ is used in Porporato-2004). According to Guswa-2010, the Porporato's solution for transpiration will lead to a slightly deeper rooting depth than the Milly's solution for transpiration (adopted in Guswa-2008 and this study). Despite that, the responses of $Z_r$ to changes in climate are essentially the same when the two transpiration solutions are adopted. Moreover, the responses of $Z_r$ to changes in $CO_2$ in the two solutions should also be essentially the same, since the effects of $CO_2$ on $Z_r$ are expressed via water use efficiency and potential transpiration in our parameterization, which are independent of $Z_r$ parameterizations. In summary, using different transpiration solutions (Milly-1993 versus Porporato-2004) would only lead to difference in the resultant absolute magnitude of runoff ($Q$) but unlikely to result in differences in the response of $Q$ to $CO_2$ changes in any notable way, especially when the relative magnitude is used.

In the revised manuscript, we have mentioned this point as a model limitation and discussed this limitation (Line 421 to 441). The relevant new text reads: "Finally, it is worthwhile noting there are several limitations in the developed modeling framework. First, Guswa's (2008) rooting depth model adopted herein employs an intensive root water uptake strategy, which assumes that root water uptake occurs at a potential rate (i.e., $E_{P\_T}$) until soil moisture reaches the wilting point when transpiration is completely suppressed (Guswa, 2008). This intensive root water uptake strategy differs from the root water uptake strategy employed in Porporato et al.'s (2004) stochastic soil water balance model, which is a more conservative strategy under which root water uptake linearly decreases with the decrease of soil moisture (Porporato et al., 2004). Combining the two strategies in one modeling framework potentially leads to inconsistency in the theoretical aspect of the approach. In fact, a later study by Guswa (2010) incorporated Porporato et al.'s (2004) soil water balance model into his cost-benefit framework for rooting depth (referred to as the Guswa-2010 approach). However, the Guswa-2010 approach could not provide an explicit solution for $Z_r$, because the solution of transpiration in Porporato's model is an incomplete gamma function of $Z_r$ (Guswa, 2010; Porporato et al., 2004). As a result, to allow an analytical solution to be derived we used Guswa (2008) for $Z_r$ in our modeling framework. According to Guswa (2010), using the conservative root water uptake strategy would result in a slightly deeper $Z_r$ compared to that if the intensive strategy were used. Despite that, the response of $Z_r$ to changes in $C_a$ under the two strategies should be similar, as the effects of $eCO_2$ on $Z_r$ are expressed via water use efficiency and $E_{P\_T}$ in our parameterization, which are independent of $Z_r$ parameterizations. This means that adopting different root water uptake strategies would only lead to differences in the resultant absolute magnitude of runoff ($Q$) but

unlikely to result in differences in the response of $Q$ to eCO$_2$, especially when the relative magnitude is used (Figures 5d, 5e and 6a, 6b, 6e and 6f)."

R1C3: Budyko curve parameterization. The authors use results from Porporato et al. (2004) to link the exponent n in Eq. (1) to rooting depth, water holding capacity, and mean precipitation event depth. This approach is based on analysis of "data from Porporato et al. (2004)" (L103), though it is important to emphasize that in that paper there are no data (except for net primary productivity), so the regression reported in Eq. (2) is obtained by fitting results from the analytical model in Porporato et al. (2004). This step is quite unnecessary, since the results are already in a close-form solution, which can be used directly without any fitting. In other words, Porporato et al. (2004) already provides a fully parameterized Budyko curve, which should be used for consistency with the other parts of the model instead of Eq. (1).

Reply: The data were obtained directly from Porporato et al. (2004) is their numerical solutions of the corresponding $E/P$ for every 0.1 increment in $E_P/P$ for the six $\omega$ curves. By numerically solving the Choudhury's formulation of the Budyko curve, we determined the values of the Budyko parameter ($n$) that correspond to the E/P values of each of the six $\omega$ curves. We then pooled all $n - \omega$ pairs and derived a simple relationship between them (Eq. 2 in the manuscript). Relevant information is now provided in the revised manuscript.

The relevant text now reads (Line 113 to 119): "A relationship between Porporato's $\omega$ parameter and Choudhury's $n$ parameter was built following three steps. Firstly, we obtained the numerical solution of the Porporato's model of the corresponding $E/P$ for every 0.1 increment in $E_P/P$ for six separate $\omega$ curves. Secondly, by numerically solving the Choudhury's formulation of the Budyko curve, we determined the values of the Choudhury's parameter ($n$) that correspond to the $E/P$ values of each of the six $\omega$ curves. Thirdly and finally, we pooled all $n - \omega$ pairs together and deduced the relationship between $n$ and $\omega$"

We agree with this reviewer from a theoretical perspective. However, from a practical perspective, Porporato's model is much more complex than the Budyko model and can only be solved numerically (for the reason stated in the reply to R1C2). With a specified model parameter, Porporato et al. (2004) proved the similarity between their solution and the Budyko's solution of mean annual water balance. Compared with Porporato et al (2004), the Budyko's formulation is much simpler, which allows an analytical attribution of $Q$ changes. Therefore, developing relationship between the Budyko's parameter (here the Choudhury's expression of the Budyko curve) and Porporato's parameter is a simple yet effective way to solve the problem. The same approach has been adopted previously (e.g., Donohue et al., 2012, https://doi.org/10.1016/j.jhydrol.2012.02.033; Liu et al., 2016, https://doi.org/10.1016/j.jhydrol.2016.10.035; Yang et al., 2016, https://doi.org/10.1002/2016WR019392; Shen et al., 2017, https://doi.org/10.1016/j.jhydrol.2017.09.023; Zhang et al., 2018,

https://doi.org/10.1002/2017WR022028).

R1C4: Model interpretation at annual time scale. The models by both Porporato et al. (2004) and Guswa (2008) have been developed for growing season conditions, assuming no seasonality in precipitation and potential evapotranspiration. In this contribution, these models are interpreted as representative of the whole hydrologic year and used to partition variability in annual runoff. I wonder if and how the original model assumptions and the current model interpretation can be reconciled.
Reply: In our study, the Guswa's model was indeed applied for growing season to determine the effective rooting depth. We have made this point explicit in the revised manuscript (Line 126). The growing season is determined following Zhu et al. (2016), which was explicitly stated in L133-134 of our revised submission (Line 117 in our original manuscript). The determined effective rooting depth during growing season is then used to determine the Porporato's parameter and further, the Budyko parameter. It should be noted that the effective rooting depth is essentially the maximum depth of the hydrologically active soil layer, which should remain unchanged between the growing season and the whole hydrologic year.

R1C5: Role of precipitation event frequency. Eq. (2) neglects the effect of precipitation event frequency on the shape of the Budyko curves from Porporato et al. (2004) framework. The variations in frequency across climates can be more pronounced than variations in mean event depth.
Reply: We agree with this reviewer that the precipitation event frequency is important in the control of partitioning precipitation into evapotranspiration and runoff and the Porporato et al (2004)'s framework does not explicitly account for it in their model parameterization. Nevertheless, the Porporato's framework considers both the total precipitation amount and the mean event depth, which when combined provide information about event frequency. Therefore, the effect of variation in event frequency on the hydrological partitioning in the Porporato's framework and the BCP model is implicitly expressed by the variations in both total precipitation amount and mean event depth.

R1C6: Interpretation of results from Donohue et al. (2013). Eq. (6) presents an iterative scheme to estimate changes in WUE through time, but in the original articles by Donohue et al. (2013, 2017) steady state models are developed, without an explicit dynamic component. The time scales to achieve steady state are probably in the order of decades (necessary for vegetation change), not years as indicated in Eq. (6).
Reply: Eq. (6) follows the gas-exchange theory at the leaf-level to quantify the response of WUE ($W$ in the following equations for simplicity) to elevated $CO_2$, originally given by Wong et al. (1979),

$$W_L = \frac{A_L}{E_{TL}} = \frac{g_s(C_a - C_i)}{1.6 g_s(v_i - v_a)} = \frac{C_a}{1.6v}(1 - \frac{C_i}{C_a})$$

where $A$ (g C m$^{-2}$ s$^{-1}$) and $E_T$ (mm s$^{-1}$) stand for the assimilation and transpiration rate,

respectively, and the subscript L denotes the leaf-level variables. $C_a$ (ppm) and $C_i$ (ppm) respectively represent the ambient and intercellular concentration of $CO_2$, and $v_a$ (Pa) and $v_i$ similarly represent ambient and intercellular concentration of water vapor while $g_s$ (m s$^{-1}$) is the stomatal conductance to $CO_2$. The numeric factor 1.6 accounts for the greater diffusivity of water vapor relative to $CO_2$ in air [Wong et al., 1979]. We use $v$ to denote the leaf-to-air water vapor pressure difference (Pa), which is approximated by the atmospheric vapor pressure deficit in subsequent analysis. The relative change in $W_L$ is given by:

$$\frac{dW_L}{W_L} = \frac{dA_L}{A_L} - \frac{dE_{TL}}{E_{TL}} = \frac{dC_a}{C_a} - \frac{dv}{v} + \frac{d(1-\frac{C_i}{C_a})}{(1-\frac{C_i}{C_a})} \quad .$$

Observations have shown that for a given photosynthetic pathway (*i.e.*, C3 or C4 species), $C_i/C_a$ is relatively conservative [Arens et al., 2000, Long et al., 2004, Wong et al., 1979]. The response of the term $1 - C_i/C_a$ to a change in $v$ can be quantified by taking $1 - C_i/C_a$ as being approximately proportional to the square root of $v$ [Donohue et al., 2013; Farquhar et al., 1993, Medlyn et al., 2011]. Therefore, Eq. (2) can be written as:

$$\frac{dW_L}{W_L} = \frac{dA_L}{A_L} - \frac{dE_{TL}}{E_{TL}} \approx \frac{dC_a}{C_a} - \frac{1}{2}\frac{dv}{v} \quad .$$

Eq. (6) in the current manuscript is essentially the same as the above equation. This equation does not require steady-state to be satisfied. However, the above equation is for leaf-level fluxes. Applying this equation at the canopy-scale implicitly assumes the same upscaling factor when converting the leaf-level assimilation and transpiration to the canopy level for a given location. This assumption is also adopted in Donohue et al. (2013, 2017). We have made this assumption explicit in the manuscript (Line xxx) and text below. It is also noted that Donohue et al. (2013, 3017) applied this equation at the same 5-year period as in the current study.

This reviewer did point out an important issue that this theory works better for undisturbed and mature vegetation but can be problematic for disturbed and immature vegetation (e.g., seedlings). However, the issue of vegetation age and disturbances is very complex and is well beyond the scope of this manuscript, noting there are no global dataset monitoring vegetation age that we could use in our modelling.

In the revised manuscript we discuss these points (Line 441-456). Relevant text reads: "The second limitation of the current study lies in the steady-state assumption of the modeling framework. More specifically, the steady-state assumption is made in (i) catchment water balance and (ii) vegetation functioning. For (i), a five-year period does not necessarily guarantee zero-storage change. Nevertheless, the imbalance in water balance calculation under a steady-state assumption at a five-year scale is generally very small (i.e., typically less than 6% of $P$ in arid regions and less than 3%

of $P$ in humid regions) (Han et al., 2020). For (ii), both the Guswa's model for $Z_r$ and Donohue's model for $L$ (see Section 2.1.5) adopted herein were developed for steady-state vegetation (i.e., mature and undisturbed vegetation). Applying these two models to immature (e.g., seedlings) and/or disturbed vegetation can be problematic because immature and/or disturbed vegetation may have very different water use and carbon allocation strategies compared to steady-state vegetation (Donohue et al., 2017; Kuczera, 1987). However, the issues of vegetation age and disturbances are extremely complex and are well beyond the scope of this manuscript. Moreover, global datasets of vegetation age and disturbances are currently lacking. In this light, our modeled response of $Q$ to eCO$_2$ should be regarded as if all vegetation were mature and undisturbed. Further efforts are needed to better quantify the age and disturbances of vegetation and to better understand the water use and carbon allocation strategies through the entire vegetation life-cycle and under various types of disturbances."

R1C7: Notation: several symbols are defined differently from the publications they are taken from, creating some confusion. For example, mean rainfall depth is denoted by alpha (not beta) in Porporato et al. (2004); rooting depth is denoted by Z_r (not Z_e) in Guswa (2008); symbol beta is used in Guswa (2008) as well, but has a different meaning; many symbols are used to define evapotranspiration and potential evapotranspiration, and not all are clearly defined (E_{P_T}, E_T, E_{P_M}, E_P); stomatal conductance is generally denoted by g_s, not C_s; symbol theta is used for volumetric soil moisture (not water holding capacity). To summarize, for readers familiar with the literature, reading this manuscript can be difficult because of the different meaning of commonly used symbols.
Reply: In the revised manuscript, we have re-defined some symbols and clearly defined all symbols to improve the flow of the manuscript. Specifically, the definitions of $E_P$, $E_T$, $E_{P\_M}$, $E_{P\_T}$ are defined upon their first appearance, respectively in Line 103, Line 137, Line 192 and Line 126. Moreover, $Z_r$ is now used for rooting depth, $g_s$ is used for stomatal conductance, WHC is used for water holding capacity and $\alpha$ is used for mean rainfall depth. $\beta$ is now used to represent a resource availability index.

R1C8: L26: why "implicitly" - do you mean "explicitly"?
Reply: Done. Revised as suggested (Line 26).

R1C9: L31: "the resource availability gradient" suggests that this gradient has been presented before, but it is not.
Reply: Thanks for pointing this issue out. We have changed "the resource availability gradient" to "a resource availability gradient" in the revised manuscript (Line 31).

R1C10: L50: other recent works have discussed these issues, including Fatichi et al. (2016, www.pnas.org/cgi/doi/10.1073/pnas.1605036113).
Reply: Done. The work of Fatichi et al., 2016 is cited in the revised manuscript (Line 50). Thanks.

R1C11: L61: please check spelling of BCP model author names.
Reply: Sorry for the typo. We have fixed it in the revised manuscript (Line 63).

R1C12: L67 and 69: are "model parameter" and "land surface parameter" indicating the same quantity?
Reply: Yes, they are the same parameter. We now use "model parameter" consistently throughout the revised manuscript to avoid potential misunderstandings.

R1C13: L81: this could be a good place for a summary of the research questions or aims of the work.
Reply: The aim of the work is summarized at the beginning of the last Introduction paragraph (Line 62-67). Putting it at the beginning of this last introductory paragraph instead of the end improves the flow of the manuscript.

R1C14: L96-97: just a comment - typically, ET is estimated from precipitation and runoff, since ET is the most difficult term in the catchment water balance to estimate; here the water balance is used to estimate Q, assuming the ET is known.
Reply: The Choudhury's formulation of the Budyko curve expresses actual ET as a function of $P$, PET and a model parameter. Then, the assumed steady-state water balance is used to calculate $Q$ as a residual (i.e., $P$ – actual ET = $Q$).

R1C5: L137: some words missing - e.g., "parameters"?
Reply: Done and thanks. Revised as suggested (Line 182).

R1C16: L139: but evaporation from the soil surface is neglected here (L91), so I am not sure I understand this statement.
Reply: We believe this reviewer misunderstood our approach. L91 (original manuscript) reads "by taking soil surface resistance equal to zero". This means that soil evaporation would occur at its potential rate.

R1C17: L147: I would define here symbols E_{P_M} and O.
Reply: Done. Revised as suggested (Line 192 to 194)

R1C18: L150: not clear how E_{P_M} differs from E_P.
Reply: In our study, $E_P$ is calculated using the Shuttleworth-Wallace two source evapotranspiration model (described in Section 2.1.2 of the revised manuscript), where $E_P$ is also affected by vegetation parameters ($L$ and $g_s$). Since both $L$ and $g_s$ respond to eCO$_2$, $E_P$ also responds to eCO$_2$. As a result, we define $E_{P\_M}$ as the component of $E_P$ that only varies with meteorological variables (or under a constant CO$_2$). This term is defined in Line 192.

R1C19: L156: this sentence is hard to follow.
Reply: We have revised relevant sentences to make it easier to follow. Relevant text

reads (Line 199 to 201): "The first term on the right hand of Eq. (23) represents $dQ$ caused by $P$ change and the second term represents $dQ$ caused by $eCO_2$. The third term calculates $dQ$ induced by changes in $E_{P\_M}$ and is calculated as $\frac{\partial Q}{\partial E_P} dE_P -$

$\frac{\partial Q}{\partial E_P} \frac{\partial E_P}{\partial C_a} dC_a$."

R1C20: L160: singular "affects".
Reply: Done. Revised as suggested (Line 204).

R1C21: Section 2.3: I would emphasize that this dataset covers experiments with artificially elevated CO2.
Reply: Done and good point. Revised as suggested. Relevant text reads: "The response of leaf-level stomatal conductance ($g_s$) response to $eCO_2$ was determined using 244 field experiments with artificially elevated $CO_2$ across a broad range of bioclimates (Ainsworth and Rogers, 2007)." (Line 213 to 215)

R1C22: L210: how was beta calculated?
Reply: The mean rainfall intensity was calculated following the IPCC Fifth Assessment Report (Hartmann et al., 2013), which is the ratio of annual total precipitation over the number of wet days (with daily precipitation higher than 1 mm). The information is now provided in the revised manuscript (Line 255 to 257). Relevant text reads: "The mean rainfall intensity was calculated as the ratio of annual total precipitation over the number of wet days (with daily precipitation higher than 1 mm; Hartmann et al., 2013)."

R1C23: L236: "differentially better" - meaning not clear.
Reply: Done. We have deleted "differentially" in the revised manuscript (Line 288).

R1C24: L238: these statements are qualitative and no performance measure is provided to compare the two model variants.
Reply: Done. We added the performance measures to compare the two modelling results in the revised manuscript (Line 287 to 296).

Relevant text reads: "Results show that the BCP model, when considering $eCO_2$, performed better in estimating $Q$ trends than the BCP model without considering $eCO_2$, as evidenced by an improvement of $R^2$ by 0.02, a reduction of RMSE by 0.03 mm yr$^{-2}$ and a decrease of mean bias by 0.11 mm yr$^{-2}$, averaged over all catchments (Figure 4d). More apparent improvements of the BCP model performance with the consideration of $eCO_2$ are found in regions having a relatively higher resource availability index. For $\beta$ of 0.4-0.6, 0.6-0.8 and 0.8-1.0, the mean bias of simulated $Q$ trends with $eCO_2$ is -0.02 mm yr$^{-2}$, 0.06 mm yr$^{-2}$, -0.36 mm yr$^{-2}$ but increased to 0.24 mm yr$^{-2}$, 0.20 mm yr$^{-2}$ and -0.53 mm yr$^{-2}$, respectively, when $eCO_2$ is not considered (Figure 4d). These results suggest that the analytical framework developed herein

captures the eCO$_2$ signal on the observed $Q$ changes."

R1C25: L249: ". . .caused an increase of L" - in the remote sensing data or based on model predictions?
Reply: This is a modelling results, since remotely sensed trends in $L$ is a combined results of different forcing factors. Here, the increases in $L$ are only driven by the CO$_2$ fertilization effect. We have made it explicit that this is a modelling result in the revised manuscript (Line 307, 311, 315 and 318).

R1C26: L252: "L increase is found. . ." - in the remote sensing data or based on model predictions?
Reply: This is a modelling result. Also see our reply to R1C25.

R1C27: L265: suggested rewording ". . . shows a slight decrease in. . ."
Reply: Done. Revised as suggested (Line 323).

R1C28: L348: I am not sure how results here can guide climate model development.
Reply: Originally, our idea, which we agree we did not articulate well, was that the response of LAI to eCO$_2$ along the resource availability gradient reported in herein (also observed by FACE) may provide some useful guidance to the development of ecosystem model, as current ecosystem models perform poorly in representing this response. In addition, the current ecosystem models do not (at least explicitly) consider the response of plant roots to changes in CO$_2$, and this response should be considered in future ecosystem model development. However, as this discussion is not concrete in the revised manuscript we deleted relevant statements about future model development thus avoiding any potential misunderstandings.

R1C29: Figure 3: please check units of RMSE and mean bias in panel (b).
Reply: Sorry for the typo. We fixed it in the revised manuscript (see Figure 4b in the revised manuscript).

R1C30: Figure 4: are the shown changes in L modelled or measured from remote sensing? Note that "but for each" in the caption is repeated.
Reply: These are modelled results. We have made it explicit in the revised captions of Figure 5. In addition, errors in the caption are fixed.

R1C31: Figure 6: I suspect L587-590 are not meant to be in the caption (they seem not relevant). I would also show error bars consistent with other plots – here they represent 1/10 of standard deviation, indicating that in fact the variance is extremely large.
Reply: Errors in the caption are corrected. This reviewer was correct that the variance is indeed very large. In the revised figure, we show one standard deviation among grid-boxes using figures instead of error bars. This helps to better compare the difference in means (showing standard deviation as error bars would make the scale

of Y-axis too large to compare the difference in means).

Thanks very much for your questions and comments that helped us improve our manuscript.

**Anonymous Referee #2**
(Black = original reviewer comments; blue = our response; and green = new or revised text).

R2C1: The manuscript by Yang et al. aims at quantifying the impact of physiological and structural vegetation adaptations induced by elevated atmospheric CO2 concentration (eCO2) on mean annual runoff (Q). The vegetation-mediated eCO2 effect on Q is complex and involved several processes with sometimes opposite effects. Also, the link of below-ground processes to eCO2 is still not entirely clear. For these reasons, the effect of eCO2 on Q is a source of uncertainty in simulation models. This paper uses an attribution framework, based on the previously applied BCP model, to quantify the net vegetation-mediated eCO2 effect on Q. This is a highly topical subject, the choice of methods seems appropriate and the inclusion of a link to below-ground processes constitutes a substantial novelty, which makes this manuscript of interest to HESS. However, my concerns relate to the presentation of the material: I find the manuscript difficult to follow and think that its value could be greatly increased by improving the description of methods. I therefore recommend a minor revision before the paper gets published.

Reply: Thanks for your favorable evaluation of our study. We respond to / address your individual comments below.

R2C2: I find the presentation of the methods somewhat unclear and found it difficult to understand how the different methodical steps are linked together, particularly Sections 2.3 and 2.4. Are the responses of stomatal closure and L to eCO2 integrated in the BCP model? If so, please make the links explicit. If not, please clarify how these different steps work together in the attribution framework. Also, it seems to me that the step of extending the analysis from the study catchments (l. 196 states that the analysis is limited to those) to a global raster map (e.g. Fig. 7) is not described in sufficient detail in the Methods.

Reply: Thanks for your suggestion. Following your comments, we have added a beginning paragraph in the Method section and Figure 1, which summarized the links between different parts of the BCP model and how the stomatal and structural responses (new section 2.1.4 and 2.1.5) are incorporated into the BCP model to determine the eCO₂ effect on $Q$.

Relevant text (Line 87-99) reads: "The Budyko-Choudhury-Porporato (BCP) model was adopted here to simulate $Q$ and to attribute changes in $Q$ (Yang et al., 2016b; Donohue et al., 2012). Briefly, the BCP model uses the Choudhury's (1999) formulation of the Budyko curve to estimate $Q$ (Eq. 1 below), in which the model parameter is estimated based on the relationship between the Choudhury's model parameter and the Porporato's model parameter (Eq. 2 below). The required rooting depth ($Z_r$) in estimating the Porporato's parameter is calculated using the Guswa's (2008) rooting depth model (Eqs. 3-5 below). To quantify the response of $Q$ to eCO₂ via vegetation feedbacks, the stomatal response of vegetation to eCO₂ is determined

by upscaling the observed response at the site level to the biome level (Section 2.1.4) and the Leaf area index ($L$) response to eCO$_2$ is quantified based on the response of *WUE* to eCO$_2$ adjusted by the local resource availability following Donohue et al. (2017) (Section 2.1.5). The effects of eCO$_2$ on both stomatal and $L$ also affect rooting depth in Guswa's (2008) model. A flowchart of our modeling approach is summarized in Figure 1 and detailed calculation procedures are described in Sections 2.1.1 to 2.1.5."

[Figure]

**Figure 1** Flowchart of using the BCP model to detect the eCO$_2$ impact on $Q$. The terminologies used are explained in the following text (section 2.1.1 through 2.1.5).

In addition, we removed the catchment scale analysis and presented the modelling results across the entire global vegetated lands in the revised manuscript (Figures 5-9). The catchment observations of $Q$ were only used for model validation in this new version of the manuscript (Figure 4). We revised relevant text to accommodate this change (e.g., Line 67).

R2C3: In the presentation of the results, it is not immediately clear if the Q-eCO2 response refers to the net effect of increased CO2 concentration on Q (through all the known effects on e.g. meteorological forcing, plant physiological and structural adaptations to CO2 and climate etc – this seems to be the case in the first paragraph of Section 3.3 and Fig. 5), or the net effect of eCO2-induced plant physiological and structural adaptations (this seems to be the case in the second paragraph of Section 3.3 and Fig. 6). Then again, in the first paragraph (l. 270 ff.) the authors discuss the relative importance of physiological and structural effects of eCO2 on vegetation before the corresponding evidence has been presented.
Reply: The reported effect does not include the CO$_2$ effect on meteorological forcing

but only for the $CO_2$ effect on $Q$ via vegetation feedbacks (plant physiological and structural responses). We have now made this explicit in the revised manuscript (Line 29, line 67, line 327 and line 373).

R2C4: The authors conclude by stating that the analyses provide insightful guidance for the development of climate models. It would be helpful to describe how exactly the findings from this analysis can be used in climate model development. In general, if this is where the value of the paper lies, it would greatly benefit from connecting the different steps (methods and discussion) to the current state of research in climate and earth system modeling (including the significance of these feedbacks and their uncertainty for earth system modeling, e.g. Hickler et al. 2015 https://doi.org/10.1007/s40725-015-0014-8, Li et al. 2019 https://doi.org/10.5194/bg-15-6909-2018) . For example, how does the CO2 fertilization effect calculated in this paper compare to results obtained in modeling studies? How is the link between Ca and below-ground vegetation dynamics currently represented in models, and how might they benefit from the advances in this study?
Reply: Originally, our idea, which we agree we did not articulate well, was that the response of LAI to $eCO_2$ along the resource availability gradient reported in herein (also observed by FACE) may provide some useful guidance to the development of ecosystem model, as current ecosystem models perform poorly in representing this response. In addition, the current ecosystem models do not (at least explicitly) consider the response of plant roots to changes in $CO_2$, and this response should be considered in future ecosystem model development. However, as this discussion is not concrete in the revised manuscript we deleted relevant statements about future model development thus avoiding any potential misunderstandings.

R2C5: l. 111: Please indicate the values for root respiration and the Q10 parameters.
Reply: In this study, we use the standard $Q10$ theory to calculate root respiration. The parameters used in the $Q10$ equation are taken from the MODIS NPP algorithm. We have added relevant information in the revised manuscript (Line 126-129).

Relevant text reads: "$\gamma_r$ is the root respiration rate (g C g$^{-1}$ roots day$^{-1}$), which is quantified using the standard $Q_{10}$ theory (Lloyd and Taylor, 1994; Ryan, 1991) with a fixed $Q_{10}$ coefficient of 2.0 (Zhao et al., 2011). The base respiration rate at 20 ºC for each biome type is determined following Heinsch (2003)."

R2C6: l. 142 ff: This is not necessarily the case. In the Guswa model, the relation of optimal rooting depth to P/EP is nonlinear and non-monotonic, with the greatest optimal depth calculated in conditions where water supply and demand are approximately equal.
Reply: Our apologies for not expressing this well in our original submission; the statement here discusses the potential $CO_2$ fertilization effect on rooting depth and not how rooting depth would respond to climate variations. We have made this point clear in the revised manuscript (Line 185).

R2C7: Eq. 15: please define beta.

Reply: As suggested by Reviewer #1 (i.e., R1C7), we used symbol $\alpha$ to represent mean rainfall intensity, which is first introduced in line 111-112 and used in Eq. 2.

R2C8: l.161: what exactly does "residual" mean in this context?

Reply: Residual here refers to total $dQ$ minus $P$-induced $dQ$ plus $E_{P\_M}$-induced $dQ$ plus eCO$_2$-induced $dQ$. In our parameterization, this other effect may include $dQ$ caused by changes in rainfall intensity and/or climate-driven (other than eCO$_2$-induced) vegetation changes. We have made this explicit in the revised manuscript (Line 205-206).

R2C9: Eq. 16: What are the units of S_Q_to_eCO2?

Reply: Apologies for our oversight; the units of $S_{Q\_to\_CO2}$ in Eq. 16 is mm yr$^{-1}$ ppm$^{-1}$. This information is added in the revised manuscript (Line 207).

R2C10: l. 250: This average value by itself is not very informative, I suggest characterizing the distribution (mode(s) and range) in more detail (including a discussion of Fig. 4 b).

Reply: We have provided mean plus one standard deviation in the revised manuscript (Line 302, 308, 316).

The discussion of Figure 4b is provided in the discussion section (Line 388-403). Relevant text reads: "The positive response of $Q$ to eCO$_2$ in high $\beta$ catchments (primarily located in tropical rainforests; Figure 6a) implies a dominant effect of eCO$_2$-induced partial stomatal closure over increases in $L$ and $Z_r$ on $E$ in these environments (Figure 6). This is reasonable, as both theoretical predictions and *in-situ* observations have consistently reported a negligible response of $L$ to eCO$_2$ in humid and closed-canopy environments (Donohue et al., 2017; Yang et al., 2016a; Norby and Zak, 2011; Körner and Arnone, 1992). In such environments, water is generally abundant with light and/or nutrient availability being the most limiting resources for vegetation growth (Nemani et al., 2003; Yang et al., 2015), and vegetation have evolved to efficiently capture light by maximizing their above-ground structure (i.e., $L$). As a result, in these high $L$ regions, vegetation have already absorbed most of the incident light and any extra leaves would not materially increase the light absorption (Yang et al., 2016a). By contrast, in dry regions, eCO$_2$-induced increase in vegetation water use efficiency (so less transpiration for the same amount of carbon assimilation at the leaf-level) would lead to an increase in $L$ that is directly proportional to an increase in water use efficiency which would increase canopy-level carbon fixation (Figure 5b). This finding is consistent with satellite observations (Donohue et al., 2013) and *in-situ* FACE experiments (Norby and Zak, 2011)."

R2C11: l. 257 "has resulted": I suggest making it clearer that this statement describes simulation results, rather than observations (as I understand it).

Reply: Done. We have made it explicit that this is a modeling result (Line 307, 311,

315 and 318).

R2C12: l.288: did you mean "other factors including"?
Reply: Done. Revised as suggested (Line 355).

R2C13: l. 320: which mechanism?
Reply: We replaced "the $Q$-e$CO_2$ response mechanism" with "the $Q$-e$CO_2$ response pattern" in the revised manuscript (Line 386). The pattern is described in the previous sentence (Line 384-385), being "a significant positive trend ($p<0.01$) in the $Q$-e$CO_2$ response along the resource availability gradient".

R2C14: l.337: I am not sure if the word "exaggerate" corresponds to the idea expressed by the authors. Maybe "exacerbate"?
Reply: Done. Revised as suggested, thanks, please see Line 408).

R2C15: l. 340 "This suggests that the structural response. . ." This causal link is not immediately clear to me, please clarify
Reply: The structural response of vegetation to e$CO_2$ decreases with the increase of leaf area index, so with the increases of leaf area index in future climate scenarios (due to higher precipitation and $CO_2$) this means that the response of vegetation structure (e.g., leaf area index) to elevated $CO_2$ will eventually decrease. This point is made more clearly in the revised manuscript (Line 411-414). Relevant text reads: "As the vegetation structural responses to e$CO_2$ decreases with the increase of $L$, the predicted future $L$ increases suggest that the structural response of vegetation to e$CO_2$ may eventually decrease and the physiological effect of vegetation to e$CO_2$ may become increasingly dominant in the overall response of vegetation water use to e$CO_2$."

R2C16: Fig. 4 a,b,d,e: To avoid any confusion I think it is important to make clear that the data shown are the results of simulations, and not (as I understand it) based on observations.
Reply: Done. Revised as suggested (see new caption of Figure 5, which was Figure 4 in our original submission).

R2C17: Fig. 5: What exactly is meant by "Q change induced by eCO2" (see my comment in the 3rd paragraph)?
Reply: The e$CO_2$ effects examined here are only via vegetation feedbacks (i.e., it does not include the e$CO_2$ impacts on meteorological forcing variables). Please also see our reply to R2C3.

R2C18: Fig. 6: The size of the error bars representing 1/10 suggests a great variability of these quantities among the different catchments. Consider using an alternative visualization method (e.g. boxplots or kernel density plots).
Reply: This reviewer was correct that the variance is indeed very large. In the revised

figures (now Figure 7f and Figure 8), we show one standard deviation among grid-cells using figures instead of error bars. This helps to better compare the difference in means (showing standard deviation as error bars would make the scale of Y-axis too large to compare the difference in means). Please see our response to R1C31.

R2C19: Fig 6: some sentence of the caption refer to elements that I cannot see (viewing the PDF in Chrome on Windows): values in parenthesis; vertical grey dashed line.
Reply: Our apologies, there were some mistakes in the caption of our original Figure 6. We have corrected these mistakes in the revised manuscript (please see Figure 8 of our revised manuscript).

Thanks very much for your review which helped us improve our manuscript.

**Anonymous Referee #3**
(Black = original reviewer comments; blue = our response; and green = new or revised text).

R3C1: The authors explore past runoff trends over undisturbed catchments and globally. Using an analytical framework, they attribute runoff trends to climate and vegetation influences along a resource availability index. The impact of CO2-induced vegetation changes on runoff has remained highly uncertain and as such, this study is a valuable contribution to the literature and well suited to HESS.
Reply: Thanks for your encouraging and constructive comments; replies to your individual comments are provided below.

R3C2: Whilst I find this study interesting, it is a shame it does not go further in quantifying the CO2-induced vegetation changes on Q. In particular, the authors mention the inclusion of CO2-induced rooting depth changes as a key novel aspect. However, whilst the authors quantify in detail the influence of CO2 on the individual above- and below-ground vegetation processes, it is not shown how these in turn affect Q. For Q, only the bulk CO2 response is presented if my reading of the results is correct. A number of studies already exist on the bulk CO2 responses and/or separating the effects of stomatal closure and LAI on Q (although I appreciate a new modelling framework is introduced here). Here it would have been interesting to know how Ze specifically changes Q. I think the results suggest the influence of rooting depth changes are minimal but this is glossed over in the discussion.
Reply: This is a very good suggestion; thanks. However, in our framework, the impacts of $eCO_2$ on vegetation structures (both LAI and rooting depth) are calculated simultaneously, making it impossible to separately calculate the LAI-induced runoff changes and rooting depth-induced runoff changes. Specifically, $eCO_2$-induced rooting depth change is parameterized through changes in WUE and potential transpiration, and the $eCO_2$-induced potential transpiration change is caused by $eCO_2$-induced changes in stomatal conductance and LAI. Hence, our framework does not allow a separate evaluation of LAI and rooting depth effects.

While not exactly your suggestion (which focused on above-ground and below-ground separation of Q impact), following your very good suggestion, we performed new analysis to examine how the physiological and structural responses of vegetation to $eCO_2$ affect runoff separately (Figure 6 and Line 326-342). Relevant text reads: "Over 1982-2010, $C_a$ increased by ~12.1%. For the same period, the BCP model detected a very small reduction in $Q$ of ~1.7% (or 2.2 mm $yr^{-1}$) induced by $eCO_2$ via vegetation feedbacks across the entire global vegetated lands (Figures 6b and 7d). This 1.7% reduction in $Q$, under the context of 12.1% increases in $C_a$, demonstrates a muted response of $Q$ to $eCO_2$. In addition, the overall negative effect of $eCO_2$ on $Q$ suggests that the structural forcing of $eCO_2$ on vegetation water consumption (both above- and below-ground) outweighs the physiological effect of $eCO_2$ driving leaflevel water saving. Across the global vegetated lands and for the same period, the physiological response of vegetation to eCO$_2$ has led to an increased $Q$ by 0.7% (or 0.9 mm yr$^{-1}$), with the simulated $Q$ increases being increasingly larger as $\beta$ increases (Figure 6d). By contrast, the structural response of vegetation to eCO$_2$ has resulted in an overall $Q$ reduction by 2.4% (or 3.1 mm yr$^{-1}$), with the decreases in $Q$ being increasingly smaller as $\beta$ increases (Figure 6e). These two opposite responses of vegetation water use to eCO$_2$ along the resource availability gradient have led to a significant positive trend ($p<0.01$) in the $Q$-eCO$_2$ response along the resource availability gradient, from a negative response in low $\beta$ landscapes to a positive response in high $\beta$ landscapes (Figure 6b). Nevertheless, an exception in found extreme arid zones (i.e., when $\beta<0.1$; Figure 6b). This is because in extremely dry areas, the availability of water defines the outcome and the sensitivity of $Q$ to any changes in land surface properties is very small (Donohue et al., 2013; Roderick et al., 2014)."

R3C3: I would also hope more clarity on how parameter n is determined. The current explanation is not sufficient, including what data were used. The methods should also be revised for clarity, reading the results it becomes unclear what was quantified using the analytical framework vs other methods (e.g. stomatal closure and L responses). Perhaps a summary of the steps at the start of Methods would help the reader.

Reply: Done. Following your suggestion, we have added a beginning paragraph in the Method section and Figure 1, which summarized the links between different parts of the BCP model and how the stomatal and structural responses (respectively new sections 2.1.4 and 2.1.5) are incorporated into the BCP model to determine the net eCO$_2$ effect on $Q$.

Relevant text (Line 87-99) reads: "The Budyko-Choudhury-Porporato (BCP) model was adopted here to simulate $Q$ and to attribute changes in $Q$ (Yang et al., 2016b; Donohue et al., 2012). Briefly, the BCP model uses the Choudhury's (1999) formulation of the Budyko curve to estimate $Q$ (Eq. 1 below), in which the model parameter is estimated based on the relationship between the Choudhury's model parameter and the Porporato's model parameter (Eq. 2 below). The required rooting depth ($Z_r$) in estimating the Porporato's parameter is calculated using the Guswa's (2008) rooting depth model (Eqs. 3-5 below). To quantify the response of $Q$ to eCO$_2$ via vegetation feedbacks, the stomatal response of vegetation to eCO$_2$ is determined by upscaling the observed response at the site level to the biome level (Section 2.1.4) and the Leaf area index ($L$) response to eCO$_2$ is quantified based on the response of $WUE$ to eCO$_2$ adjusted by the local resource availability following Donohue et al. (2017) (Section 2.1.5). The effects of eCO$_2$ on both stomatal and $L$ also affect rooting depth in Guswa's (2008) model. A flowchart of our modeling approach is summarized in Figure 1 and detailed calculation procedures are described in Sections 2.1.1 to 2.1.5."

[Figure]

**Figure 1** Flowchart of using the BCP model to detect the eCO$_2$ impact on $Q$. The terminologies used are explained in the following text (section 2.1.1 through 2.1.5).

R3C4: Title: The study period doesn't cover the last three decades.
Reply: To be explicit regarding the study period we have changed the title to be "Low and contrasting impacts of vegetation CO$_2$ fertilization on global terrestrial runoff over 1982-2010: Accounting for above- and below-ground vegetation-CO$_2$ effects"

R3C5: L23-26: This sentence could be written more clearly.
Reply: Done. Revised as suggested, the relevant text reads (Line 23-26): "This is partly due to eCO$_2$-induced changes in vegetation water use being opposing at the leaf-scale (i.e., water-saving caused by partially stomatal closure) and the canopy-scale (i.e., water-consuming induced by foliage cover increase), leading to highly debated conclusions among existing studies."

R3C6: L30-34: This sentence should also be broken up into two for clarity.
Reply: Done. Revised as suggested, the relevant text reads (Line 30-34): "Globally, we detect a very small decrease of $Q$ induced by eCO$_2$ during 1982-2010 (-1.7%). Locally, we find a positive trend ($p<0.01$) in the $Q$-eCO$_2$ response along a resource availability ($\beta$) gradient. Specifically, the $Q$-eCO$_2$ response is found to be negative (i.e., eCO$_2$ reduces $Q$) in low $\beta$ regions (typically dry and/or cold) and gradually changes to a positive response (i.e., eCO$_2$ increases $Q$) in high $\beta$ areas (typically warm and humid)."

R3C7: L34: highlights -> highlight
Reply: Done. Revised as suggested (Line 35).

R3C8: L38: Suggest replacing "becoming" with "and representing"

Reply: Done. Revised as suggested (Line 38).

R3C9: L44: I would suggest Donohue is not an appropriate reference here, it is not a leafscale study.
Reply: Done. The reference "Donohue et al., 2013" is removed here.

R3C10: L50: I think the authors need to unpack this sentence a little. Many of these studies look at the net response on Q so I'm not sure what the authors mean by "different aspects"? I would argue the main reason for the discrepancies across studies is due to the different processes and assumptions included in the models. Also Ukkola et al. is not a modelling study but based on observations (similarly Trancoso et al. 2017 which should also be cited for an observational analysis). Some model evaluation has also been conducted specifically for CO2 impacts (e.g. Ukkola et al. 2016, Environmental Research Letters for a DGVM and multiple FACE papers), here the evaluation seems to be limited to the overall Q trends which is not new. A more accurate statement would be that observational and evaluation studies for CO2 effects remain limited, particularly at the regional to global scale.
Reply: Done. Revised as suggested (Line 51-53). Relevant text reads: "Moreover, observational and evaluation studies for $eCO_2$ effects remain limited, particularly at regional to global scales."

Trancoso et al. 2017 is now cited in the revised manuscript.

R3C11: L53, L265: please fix grammar.
Reply: Done. Increases -> increasing (Line 55).

R3C12: L61: should be Budyko-Choudhury-Porporato
Reply: Done. Revised as suggested (Line 63).

R3C13: L94: I think Milly and Dunne actually found that the energy-only PET best produced non-water-stressed ET from climate models (their Figure 3, associated text and conclusions). Also climate models do not simulate potential evapotranspiration so perhaps best to avoid that terminology here?
Reply: Milly and Dunne (2016) also used the two-source Shuttleworth-Wallace model to examine the effect of $eCO_2$ on vegetation stomatal conductance and the consequent impact on potential evapotranspiration (the actual evapotranspiration when water is not limiting) (their Figure 2 and Figure 3). The calculation of PET using the two-source Shuttleworth-Wallace model in our study follows Milly and Dunne (2016; https://doi.org/10.1038/nclimate3046). Yang et al. (2019; https://doi.org/10.1038/s41558-018-0361-0) further demonstrated that why Milly and Dunne found energy-only potential evapotranspiration works well is because that the warming-induced PET increases are almost entirely offset by the $eCO_2$-induced PET decreases. By using the two-source Shuttleworth-Wallace model, we are able to explicitly consider the impacts of leaf area index and conductance changes on PET.

We agree that the climate models do not simulate PET. We revised this to be "evapotranspiration estimates under non-water-limited conditions" in the revised manuscript (Line 170-171).

R3C14: L103: Could you be more specific here? Taking what data?
Reply: The data obtained from the authors of that paper is their numerical solutions of the corresponding $E/P$ for every 0.1 increment in $E_P/P$ for six separate $\omega$ curves. By numerically solving the Choudhury's formulation of the Budyko curve, we determined the values of the Budyko parameter ($n$) that correspond to the E/P values of each of the six $\omega$ curves. We then pooled all $n - \omega$ pairs together to derived a simple relationship between them (Eq. 2 in the manuscript).

Relevant text now reads (Line 113-119): "A relationship between Porporato's $\omega$ parameter and Choudhury's $n$ parameter was built following three steps. Firstly, we obtained the numerical solution of the Porporato's model of the corresponding $E/P$ for every 0.1 increment in $E_P/P$ for six separate $\omega$ curves. Secondly, by numerically solving the Choudhury's formulation of the Budyko curve, we determined the values of the Choudhury's parameter ($n$) that correspond to the $E/P$ values of each of the six $\omega$ curves. Thirdly and finally, we pooled all $n - \omega$ pairs together and deduced the relationship between $n$ and $\omega$ ($R^2$=0.96, $p$<0.001; Supplementary Figure S1):"

R3C15: L111: Not clear to me how potential transpiration is determined?
Reply: The details of $E_P$ (and its two components, potential evaporation and potential transpiration) calculation using the Shuttleworth-Wallace model are now provided in more detail in Section 2.1.2.

Section 2.1.2 reads: "

**2.1.2 The Shuttleworth-Wallace model**

The Shuttleworth-Wallace two-source evapotranspiration model (the S-W model) was used to estimate $E_P$ and its two components (potential evaporation, $E_{P\_S}$ and potential transpiration, $E_{P\_T}$) (Shuttleworth and Wallace, 1985). The S-W model estimates evapotranspiration as:

$$\lambda E_P = \lambda E_{P\_T} + \lambda E_{P\_S} = C_T PM_T + C_S PM_S \tag{7}$$

$$PM_T = \frac{\Delta A + (\rho c_p v - \Delta r_a^c A_s)/(r_a^a + r_a^c)}{\Delta + \gamma[1 + r_s^c/(r_a^a + r_a^c)]} \tag{8}$$

$$PM_S = \frac{\Delta A + [\rho c_p v - \Delta r_a^s (A - A_s)]/(r_a^a + r_a^s)}{\Delta + \gamma[1 + r_s^s/(r_a^a + r_s^c)]} \tag{9}$$

$$C_T = [1 + R_c R_a / R_s (R_c + R_a)]^{-1} \tag{10}$$

$$C_S = [1 + R_s R_a / R_c (R_s + R_a)]^{-1} \tag{11}$$

$$R_a = (\Delta + \gamma) r_a^a \tag{12}$$

$$R_s = (\Delta + \gamma) r_a^s + \gamma r_s^s \tag{13}$$

$$R_c = (\Delta + \gamma) r_a^c + \gamma r_s^c \tag{14}$$

where $\lambda$ is the latent heat for vaporization (MJ kg$^{-1}$), $\Delta$ is the gradient of the saturation vapor pressure with respect to temperature (kPa K$^{-1}$), $\rho$ is the air density (kg m$^{-3}$), $c_P$ is the specific heat of air at constant pressure (MJ kg$^{-1}$ K$^{-1}$), $\gamma$ is the psychrometric constant (kPa K$^{-1}$). $r_a^a$, $r_a^c$ and $r_a^s$ are the aerodynamic resistance (s m$^{-1}$) to heat and vapor transfer between the canopy-air space and the atmosphere, between the leaf and the canopy-air space, and between the soil surface and the canopy-air space, respectively. These three aerodynamic resistance terms are estimated following Sánchez et al. (2008). $r_s^s$ and $r_s^c$ are soil surface resistance and stomatal resistance (the reciprocal of stomatal conductance), respectively. To estimate $E_P$ using the S-W model, $r_s^s$ is set to zero and $r_s^c$ is set to its non-water stressed value (Milly and Dunne, 2016). The non-water stressed values of $r_s^c$ for each biome type are provided in Mu et al. (2007). $A$ is the available energy (equals to net radiation minus ground heat flux, W m$^{-2}$) and $A_s$ is the available energy at the soil surface, which is estimated as a function of $L$ following Beer's law (Campbell and Norman, 1998; Yang and Shang, 2013). As a result, $A - A_s$ is the available energy absorbed by the plant canopy. The impacts of eCO$_2$ on $E_P$ and its two components are obtained by allowing $L$ and $r_s^c$ to vary with $C_a$. Recently, Milly and Dunne (2016) showed that the S-W model could most satisfactorily reproduce evapotranspiration estimates under non-water-limited conditions from climate models under eCO$_2$."

R3C16: L119: I don't see where the Earth System Models are described? Also why were ESMs used rather than something more observationally constrained? Given such a short time period is taken and coupled models have their own interannual variability, taking a mean across models over such a short time period is likely to be spurious. Why wasn't observationally-driven products used, e.g. GLEAM or the TRENDY ensemble? These are of course also models but at least driven by observed meteorology

Reply: We are sorry that we missed the description of the ecosystem models used in this study. We've added relevant descriptions in the revised manuscript (Line 253-263). Relevant text reads: "To obtain a spatial pattern of *WUE*, global monthly GPP and $E_T$ estimates over 1982-1985 were obtained from 8 ecosystem models from MsTMIP (Huntzinger et al., 2013), including: (i) CLM (Mao et al., 2012); (ii) CLM4-VIC (Li et al., 2011); (iii) ISAM (Jain et al., 1996); (iv) TRIPLEX (Peng et al., 2002); (v) LPJ-wsl (Sitch et al., 2003); (vi) ORCHIDEE-LSCE (Krinner et al., 2005); (vii)

SiBCASA (Schaefer et al., 2008); and (viii) VISIT (Ito, 2010).”

MsTMIP is a model comparison program, which is similar to TRENDY. All models participated in MsTMIP are forced by observed meteorological variables. Noting that the same modelling outputs of these ecosystem models contributed to both MsTMIP and TRENDY.”

R3C17: L138-139: Why do these quantities impact Ep? Most PET estimators are mainly atmosphere-driven so if this is not the case with Shuttleworth and Wallace, more details on its calculation need to be provided for clarity.
Reply: The details of $E_P$ calculation using the Shuttleworth-Wallace model are provided in Section 2.1.2 (please see our reply to R3C15). In the Shuttleworth-Wallace model, PET is also a function of vegetation parameters (leaf area index and stomatal conductance). Using the Shuttleworth-Wallace model allows to quantify the effect of $eCO_2$ on PET via vegetation feedbacks. If PET was an 'open water PET' then we would not be able to quantify the effect of $eCO_2$ on $E_P$ via vegetation feedbacks.

R3C18: Equation 12: should the notation be f() instead of g()?
Reply: Using f() and g() indicate that the two functional relationships are different. That is, they are not the same function, hence it is appropriate scientific protocol to use different letters to denote these relationships.

R3C19: L218: Which years were used?
Reply: The year 2001 was used. This is now made clear in the revised manuscript (Line 268).

R3C20: L220: You should provide the name of the dataset (i.e. ISLSCP etc.)
Reply: Done. Revised as suggested the relevant text reads (Line 269-272): “The global C4 vegetation fraction was obtained from the International Satellite Land Surface Climatology Project (ISLSCP) Initiative II C4 vegetation percentage dataset (Still et al., 2009; http://webmap.ornl.gov/ogcdown/dataset.jsp?ds_id=932).”

R3C21: Figure 1: White regions in the map that do not match the colour scale (e.g. Greenland). Should say if/why these were masked out

Reply: Done. We have revised the caption of Figure 2 (original Figure 1) as: “**Figure 2** Spatial distributions of (a) resource availability index categories and (b) climate aridity zones over global vegetated lands for 1982-2010. For the land surface blank areas are non-vegetated regions. Respectively there are 2536, 8194, 10316, 12930 and 9093 $0.5° \times 0.5°$ resolution grid-cells in the 0.0-0.2, 0.2-0.4, 0.4-0.6, 0.6-0.8 and 0.8-1.0 resource availability index categories.”

R3C22: L232, L234: missing full stop

Reply: Done. Revised as suggested.

R3C23: L243: Please avoid using brackets like this, it is very hard to read. Suggest: with the largest Cs reduction found in C4 crops and lowest in shrubs.
Reply: Done. Revised as suggested and following R1C7, we now used "$g_s$" to represent stomatal conductance. Revised text reads (Line 300-301): "All those field experiments report a reduction of $g_s$ in response to eCO$_2$, with the largest $g_s$ reduction found in C4 crops and lowest in shrubs for the same level of eCO$_2$."

R3C24: L249: How was the Ca effect on L estimated? I'm assuming using equation 18 but it has two factors influencing L (Ca and v)
Reply: This reviewer was correct that both $C_a$ and $v$ are considered in our estimation of how $L$ respond to changes in $C_a$. The reason that $L$ increases with elevated CO$_2$ is because that WUE increases with elevated CO$_2$. However, the response of WUE to elevated CO$_2$ is also mediated by changes in $v$. In our modelling framework, we firstly estimate how WUE changes and then how $L$ respond to changes in WUE. In the revised manuscript, we have stated that "The response of $L$ to eCO$_2$ was predicted based on the response of *WUE* to eCO$_2$ adjusted by the local resource availability." (Line 220-221).

R3C25: Figure 5: Would be useful to see the spatial distribution of catchment trends. Suggest adding a map of the catchments eCO2-induced trends as an additional panel
Reply: In the revised manuscript, we removed all catchments analyses (other than the catchment validation that was originally Figure 3 and is now Figure 4) and applied our attribution framework to the entire global vegetated lands (also see our reply to R3C28). The spatial distribution of eCO$_2$-induced $Q$ changes is now shown in Figure 6a (in relative units) and 7d (in absolute units).

R3C26: Figure 6: Last panel please adjust scale to show full error bars. Also please check caption, from L587 it mentions numbers that I don't see presented in the figure
Reply: Errors in the caption are corrected. In the revised figure, we show one standard deviation among grid-boxes using figures instead of error bars. This helps to better compare the difference in means (showing standard deviation as error bars would make the scale of Y-axis too large to compare the difference in means).

R3C27: L279: I'm confused why this result differs from the number on L269 and how the changes in Q described here differ from the previous paragraph?
Reply: Oops! Sorry for the typo. The change should be -2.3 mm yr$^{-1}$.
In the revised manuscript, since we conducted our analysis across the entire vegetated lands, this value changes to -2.2 mm yr$^{-1}$ (Line 327).

R3C28: L286: I'm also confused that you have suddenly moved to global results (Fig 7). In the methods, you state that the analysis is restricted to the ~2000 catchments (L195). The text doesn't also make this transition obvious.

Reply: In the revised manuscript, we conducted our analysis across the entire global vegetated lands, and only used the $Q$ data from the 2,268 catchments to validate the modelled $Q$. The issue of transition from catchment to the entire globe is not relevant anymore. We revised relevant text to accommodate this change (e.g., Line 67).

R3C29: L292: Given alpha is determined from LAI, surely low-alpha regions can be either dry or cold?
Reply: Yes, this reviewer was correct. We have made it clear in the revised manuscript (Line 82-84). Relevant text reads: "Resource availability is typically low in dry (and/or cold) environments and increases as the climate becomes more humid"

R3C30: L348: How exactly can this framework guide model development? Firstly, the results from this study are very much in line with existing studies so no particularly novel insights are revealed. And secondly, how is this framework to help climate model development exactly? And finally, this is ultimately simply another model result. Overall this feels like a bit of a throw-away statement to try and boost the value of paper.
Reply: Originally, our idea, which we agree we did not articulate well, was that the response of LAI to eCO$_2$ along the resource availability gradient reported in herein (also observed by FACE) may provide some useful guidance to the development of ecosystem model, as current ecosystem models perform poorly in representing this response. In addition, the current ecosystem models do not (at least explicitly) consider the response of plant roots to changes in CO$_2$, and this response should be considered in future ecosystem model development. However, as this discussion is not concrete in the revised manuscript we deleted relevant statements about future model development thus avoiding any potential misunderstandings.

We do not agree with this reviewer on the assessment of "the results from this study are very much in line with existing studies so no particularly novel insights are revealed." The current study firstly shows the opposite overall effect of eCO$_2$ on $Q$ via vegetation feedback between low resource availability regions (or low LAI regions, typically dry and/or cold) and high resource availability regions (or high LAI regions, typically warm and wet). Importantly our manuscript is the first study to explicitly acknowledge that eCO$_2$ change impacts below-ground vegetation components and uses this with above-ground impacts to assess $Q$ characteristics.

R3C31: L351: Are all the datasets publicly available?
Reply: Yes, all datasets are publicly available. All data sources are provided in the Data section of the main text.

Thanks for your careful review of our manuscript, addressing your comments has resulted in an improved manuscript.

---

## Author Response (AR2)

**Response to Reviewers' comments**

We greatly appreciate the reviewers providing valuable and constructive comments on our manuscript HESS-2020-548. We seriously considered each comment and revised/improved the manuscript accordingly. The individual comments are replied below.

**Editor:**
(Black = original reviewer comments; blue = our response; and green = new or revised text).
I think the suggestions by Reviewer 2 is valuable, and suitable to improve the quality of the manuscript. I recommend that you consider them sincerely. As the reviewer at the same time assumes that the results would not change substantially, (1) I would leave it up to you whether you want to implement the proposed change of methods to enhance the consistency of the approach or add a section to the discussion about those potential improvements to the method. Similarly, (2) I agree with the reviewer on adding discussion on the effect of seasonality on the results.

Reply: Thank you very much for continuing to handle our manuscript and for providing overview suggestions. We have revised the manuscript following yours and the reviewers' comments. As for the two points you specifically mentioned, which we have numbered above to aid communication.

First, we choose not to re-perform all calculations using the new modelling framework and as per your advice mentioned it in the discussion and pointed it out as a potential future research direction, please see L447 to L450 in our revised manuscript. The scientific justification for this choice is stated in our reply to R1C1.

Line 447-450: "Alternatively, Rodríguez-Iturbe and Porporato (2004) incorporated the intensive root water uptake strategy into a stochastic soil water balance model and obtained a steady-state solution that has a simper form than Porporato et al.'s (2004) and also mimics the Budyko curve. This approach deserves further investigation."

Second, for the "growing season versus annual" issue, we have provided a discussion in the revised manuscript (please see Line 450-458 is our revised manuscript).

Line 450-458: "Second, if interpreted strictly from a theoretical perspective, Porporato et al's (2004) model is more suitable to estimate hydrological partitioning during growing seasons instead of over the entire year as it assumes a constant evaporative demand and precipitation regimes and does not account for snow

processes. Expanding all these simplifications, acknowledging imperfect knowledge and parameterisation, would require further analyses to better understand how they might affect the results shown here. Nevertheless, the uncertainties caused by these simplifications in Porporato et al's (2004) model might be partly overcome during the empirical connection made here between the Porporato's model and the Choudhury's formulation of the Budyko curve, as evidenced by the overall good performance of the developed BCP model in capturing the observed $Q$ (Figure 4)."

**Anonymous Referee #1**
(Black = original reviewer comments; blue = our response; and green = new or revised text).
R1C1: The authors responded to my comments in the first round of review, and amended the text accordingly, but I am not convinced their approach is theoretically sound (though it might work in practice). Mixing and matching analytical models with different assumptions and re-parameterizing an analytical solution using an empirical (albeit simpler) equation makes the overall approach of this contribution not very appealing nor elegant. I realize that formal elegance might not be the goal here, but there are objective shortcomings too. For example, approximating Porporato's analytical solution for the Budyko curve with Choudhury's empirical formulation results in two types of errors: first, the empirical formulation is not necessarily a good approximation, and second, the relationship between the 'n' parameter and physically-based parameters in Porporato's analytical solution also implies approximations and errors. I realize these errors are probably small, but still, it is not a 'clean' approach in my view.

The authors explain that "it is almost impossible to derive an explicit solution for Zr" for Guswa 2010 model, and that "Porporato's model is much more complex than the Budyko model and can only be solved numerically". However, the change in Q (Eq. 23) and the sensitivity S (Eq. 24) are calculated numerically anyway (L209), so there is no apparent disadvantage having a more complex solution for the Budyko curve. I do see the point that the approach adopted here has been also used in many previous papers, but that does not imply that the approach is set in stone - it can indeed be improved. Instead I agree there is a clear advantage in having an explicit solution for the rooting depth.

To sum up these thoughts, I would suggest trying the following:
1) Use Guswa 2008 model (as done in the original manuscript and in the current revision)

2) Use the Budyko curve obtained from the long-term mean water balance from the same.hydrologic model as in Guswa 2008. Essentially, it's a simpler version of Porporato 2004 already derived previously (Milly, 1993), and was also compared to Porporato 2004 in Rodriguez-Iturbe and Porporato (2004, Section 2.6.2 Minimalistic models of soil moisture dynamics); this would allow a consistent model approach throughout.

3) With this simpler Budyko curve solution, it would not be necessary to approximate the analytical solution with Choudhury's empirical formulation.

My impression is that this approach would make this work more theoretically sound, even though the main results might not be severely impacted.

Reply: Thanks for the constructive comment. However, using the suggested modelling framework deserves a new study itself, which is beyond the scope of this study. In addition, as commented by this reviewer (R1C2), the suggested modelling framework also suffers from the "annual versus growing season" issue, while this issue may be partly overcome in our modelling framework by empirically linking the Porporato's model (growing season) and the Choudhury's model (mean annual). This point is discussed in the revised manuscript (also see our reply to R1C2). In the revised manuscript, we have pointed out the Rodriguez-Iturbe and Porporato (2004)'s derivation as a potential future research direction (Line 447-450). Thanks for the suggestion.

R1C2: Regarding my comment on seasonality, the authors commented that "The determined effective rooting depth during growing season is then used to determine the Porporato's parameter and further, the Budyko parameter". I understand that the rooting depth is calculated using growing season data, but the analytical solution of the Budyko curve by Porporato 2004 is then applied to determine annual discharge. Porporato's model is not suitable to calculate annual averages if interpreted strictly - it assumes constant evaporative demand and precipitation regimes, and does not account for snow accumulation. To be strict, it should be applied to a-seasonal climates only. This requires some clarifications and discussions.

Reply: Done. This point is discussed in the revised manuscript. Relevant text reads (Line 450-458): "Second, if interpreted strictly from a theoretical perspective, Porporato et al's (2004) model is more suitable to estimate hydrological partitioning during growing seasons instead of over the entire year as it assumes a constant evaporative demand and precipitation regimes and does not account for snow processes. Expanding all these simplifications, acknowledging imperfect knowledge and parameterisation, would require further analyses to better understand how they

might affect the results shown here. Nevertheless, the uncertainties caused by these simplifications in Porporato et al's (2004) model might be partly overcome during the empirical connection made here between the Porporato's model and the Choudhury's formulation of the Budyko curve, as evidenced by the overall good performance of the developed BCP model in capturing the observed $Q$ (Figure 4)."

Minor comments

R1C3: L82: I would mention that by "resource" it is actually meant water, since as explained elsewhere light- and nutrient-limited environments have high values of resource availability according to Fig. 2.
Reply: We have indicated that resource availability is typically low in dry (and/or cold) environments and increases as the climate becomes more humid in the manuscript (Line 82-84)

R1C4: L95: acronym WUE is not defined.
Reply. Done and thanks. Revised as suggested (Line 95).

R1C5: L193: "factors… encode" (plural)
Reply. Done. Revised as suggested (Line 195).

R1C6: L205: "factors driving…"
Reply. Done. Revised as suggested (Line 208).

R1C7: L357: "factors… dominate changes…"
Reply. Done. Revised as suggested (Line 362).

**Anonymous Referee #2**

(Black = original reviewer comments; blue = our response; and green = new or revised text).

R2C1: The manuscript is much improved for clarity and the flowchart describing the methods is particularly useful. I recommend the manuscript for publication following a few minor revisions, detailed below.

Reply: Thanks for your encouraging comments. Your suggestions are replied to below and the manuscript has been revised accordingly.

R2C2: L26: Some models, including Earth System Models, used in past studies do increase root biomass in response to elevated CO2. However, none of them adjust rooting depth to my knowledge. The sentence would be more accurate stating that rooting depth is not taken into account, rather than CO2 effects on below-ground vegetation (which encompasses things like root biomass).

Reply: Done. Revised as suggested. Relevant text now reads (Line 26-27): "In addition, none of the existing studies explicitly account for $eCO_2$-induced changes to plant rooting depth."

R2C3: L172: Revise heading

Reply: Done. The heading is corrected (Line 173).

R2C4: L308: The text should reflect the fact that the uncertainty encompasses zero, i.e. no significant effect is detected

Reply: In most of the world, WUE (and thus LAI) increases with $CO_2$ unless there is an even larger increase in vapor pressure deficit. To avoid potential misunderstanding, we revised the relevant text and now also provide the 5% and 95% percentiles (Line 311 & 319).

R2C5: L319: experimental observations -> experiments

Reply. Done. Revised as suggested (Line 322).

R2C6: L356: to dominant -> to dominate

Reply. Done and thanks. Revised as suggested (Line 362).

R2C7: L357: are extensively examined -> have been extensively examined

Reply. Done. Revised as suggested (Line 363).

R2C8: L378: perhaps worth pointing out that the small CO2 effect is partly due to the two opposing processes (structural vs physiological)

Reply: Done. Revised as suggested. Relevant text reads (Line 384-385): "partly due to the two opposing water effects between the structural and physiological responses to eCO$_2$"

R2C9: L415: Would point out that many climate models only simulated physiological, and not structural changes up until recently. Hence they will inevitably simulate a net increase in Q due to CO2. In CMIP5 about half of the models used fixed LAI (and some models continue to do so in CMIP6)

Reply: Done. This point is added in the revised manuscript. Relevant text reads (Line 423-424): "Nevertheless, this may partly be because only some climate models consider the physiological effect while ignoring structural responses of vegetation to eCO$_2$."